# The LINC complex transmits integrin-dependent tension to the nuclear lamina and represses epidermal differentiation

Emma Carley[1†], Rachel M Stewart[1†], Abigail Zieman[2†], Iman Jalilian[1], Diane E King[3], Amanda Zubek[4], Samantha Lin[2], Valerie Horsley[2,4*], Megan C King[1,2*]

[1]Department of Cell Biology, Yale School of Medicine, New Haven, United States; [2]Department of Molecular, Cell and Developmental Biology, Yale University, New Haven, United States; [3]Sunnycrest Bioinformatics, Flemington, United States; [4]Department of Dermatology, Yale School of Medicine, New Haven, United States

**Abstract** While the mechanisms by which chemical signals control cell fate have been well studied, the impact of mechanical inputs on cell fate decisions is not well understood. Here, using the well-defined system of keratinocyte differentiation in the skin, we examine whether and how direct force transmission to the nucleus regulates epidermal cell fate. Using a molecular biosensor, we find that tension on the nucleus through linker of nucleoskeleton and cytoskeleton (LINC) complexes requires integrin engagement in undifferentiated epidermal stem cells and is released during differentiation concomitant with decreased tension on A-type lamins. LINC complex ablation in mice reveals that LINC complexes are required to repress epidermal differentiation in vivo and in vitro and influence accessibility of epidermal differentiation genes, suggesting that force transduction from engaged integrins to the nucleus plays a role in maintaining keratinocyte progenitors. This work reveals a direct mechanotransduction pathway capable of relaying adhesion-specific signals to regulate cell fate.

*For correspondence:
valerie.horsley@yale.edu (VH);
megan.king@yale.edu (MCK)

[†]These authors contributed equally to this work

## Introduction

Physical forces and the architecture of the extracellular environment are an emerging area of cell fate regulation (*Discher et al., 2009*). Cells sense changes in the physical environment through cell-cell and cell-extracellular matrix adhesions, providing a mechanism by which mechanical inputs can contribute to the control of cell fate. While several studies have linked geometrical and physical inputs to changes in cellular signaling through mechanoresponsive transcription factors such as YAP/TAZ (*Totaro et al., 2018*) or MKL/MRTF (*Connelly et al., 2010*), whether direct transmission of mechanical force to the nucleus can influence cell fate is not known.

The interfollicular epidermis is an excellent model to explore how mechanical inputs regulate cell fate. Epidermal stem cells adhere to the underlying basal lamina through β1 integrin-based adhesions and β4 integrin-based hemidesmosomes (*Raghavan et al., 2000*), which maintain stem cell fate, proliferation, and inhibit differentiation (*Hotchin et al., 1995*; *Levy et al., 2000*; *Rippa et al., 2013*; *Watt et al., 1993*). Upon differentiation, basal keratinocytes release their integrin-based adhesions, initiate gene expression changes to activate terminal differentiation (*Tsuruta et al., 2011*), and move upward to enter the stratified epithelium until they are shed at the skin's surface. Despite the well-established role for integrin adhesions in regulating epidermal differentiation, how integrin signals are propagated to the nucleus to regulate the keratinocyte differentiation program is not well understood.

Here, we explore whether mechanical cues can regulate keratinocyte cell fate via tension on the nucleus via the linker of nucleoskeleton and cytoskeleton (LINC) complex. The LINC complex, composed of tail-anchored Nesprins integrated into the outer nuclear membrane and SUN proteins integrated into the inner nuclear membrane, spans the nuclear envelope to mechanically integrate the cytoplasmic cytoskeleton and the nuclear interior – specifically the nuclear lamina and its associated chromatin (*Chang et al., 2015*). While the LINC complex has been postulated to act either as a direct mechanosensor or as a conduit for mechanotransduction to influence gene expression (*Alam et al., 2016*; *Wang et al., 2009*), whether the LINC complex controls gene expression in vivo remains largely untested. Indeed, to date direct gene targets of the LINC complex that regulate genetic programs in vivo, for example, during differentiation, remain to be identified.

Several lines of evidence suggest that epidermal differentiation is regulated by the nuclear lamina, yet the mechanisms remain unknown. First, during basal stem cell differentiation, a large chromosomal region termed the epidermal differentiation complex (EDC), which consists of 60 consecutive genes necessary for epidermal stratification and the production of the cornified envelope, relocates away from the nuclear lamina toward the nuclear interior to be transcriptionally activated (*Gdula et al., 2013*; *Mardaryev et al., 2014*; *Williams et al., 2002*). Further, the AP-1 transcription factor complex, itself regulated by A-type lamins, influences EDC gene expression in both proliferating and differentiating keratinocytes in vitro (*Oh et al., 2014*). AP-1 also coordinates with EZH2 in the polycomb complex, which promotes targeting of chromatin to the nuclear lamina (*Harr et al., 2015*) and regulates epidermal differentiation (*Ezhkova et al., 2009*). Lastly, a skin-specific lamin-null mouse model (Lamin B1/B2/A/C triple-knockout) exhibits a thickened epidermis attributed to precocious differentiation (*Jung et al., 2014*).

Here, we provide evidence that the LINC complex regulates epidermal differentiation in vitro and in vivo. Using a novel molecular biosensor to measure forces exerted on LINC complex molecules, we find that tension is high on the LINC complex and A-type lamins in epidermal stem cells in an integrin-dependent manner and is reduced upon differentiation. In mouse keratinocytes (MKCs) lacking LINC complexes, we observe precocious differentiation in vitro and expansion of the differentiated, suprabasal layers of the skin in vivo. This work suggests that tension from integrins is communicated through the LINC complex to the nuclear lamina to maintain the progenitor state of basal keratinocytes, providing a potential mechanism by which cell fate is regulated directly by mechanical cues.

## Results and discussion

### A tension sensor in Nesprin-2 is sensitive to integrin engagement

In order to visualize tension on the LINC complex in living cells, we generated a molecular biosensor to measure force on individual Nesprin proteins. To this end, we inserted a tension sensor module composed of the fluorescence resonance energy transfer (FRET) pair mTFP and Venus connected by an elastic flagelliform linker (*Grashoff et al., 2010*) into the juxtamembrane region of a mini-Nesprin-2 construct (*Luxton et al., 2011*), called hereafter 'N2G-JM-TSMod'. The mTFP-Venus TSMod has been shown previously to be sensitive to forces in the single pN range (*Grashoff et al., 2010*). In the N2G-JM-TSMod construct, the TSMod lies between the transmembrane domain and the entire cytosolic domain of mini-Nesprin-2, which contains both the N-terminal calponin homology domains that engage actin and a portion of the spectrin repeat region (*Figure 1A*). This construct is distinct from previously described Nesprin tension sensors in its design (*Arsenovic et al., 2016*; *Déjardin et al., 2020*). We also generated a tension-insensitive construct by inserting the TSMod at the N-terminus of mini-Nesprin2 (*Figure 1A*, 'NoT_TSMod') as well as 'dark' controls that allow for bleed through between fluorescence channels to be assessed (*Figure 1—figure supplement 1*).

We transfected primary undifferentiated MKCs cultured on fibronectin-coated glass bottom dishes with either the N2G-JM-TSMod or NoT_TSMod construct; both were targeted efficiently to the nuclear envelope when expressed at moderate levels (*Figure 1B*, D, F, H, J). We anticipated that the tension-insensitive NoT_TSMod would exhibit constitutively high FRET indexes, reflecting the conformational freedom at the N-terminus of the mini-Nesprin-2 protein. By contrast, we expected that intramolecular tension exerted on the N2G-JM-TSMod would lead to a decrease in the FRET

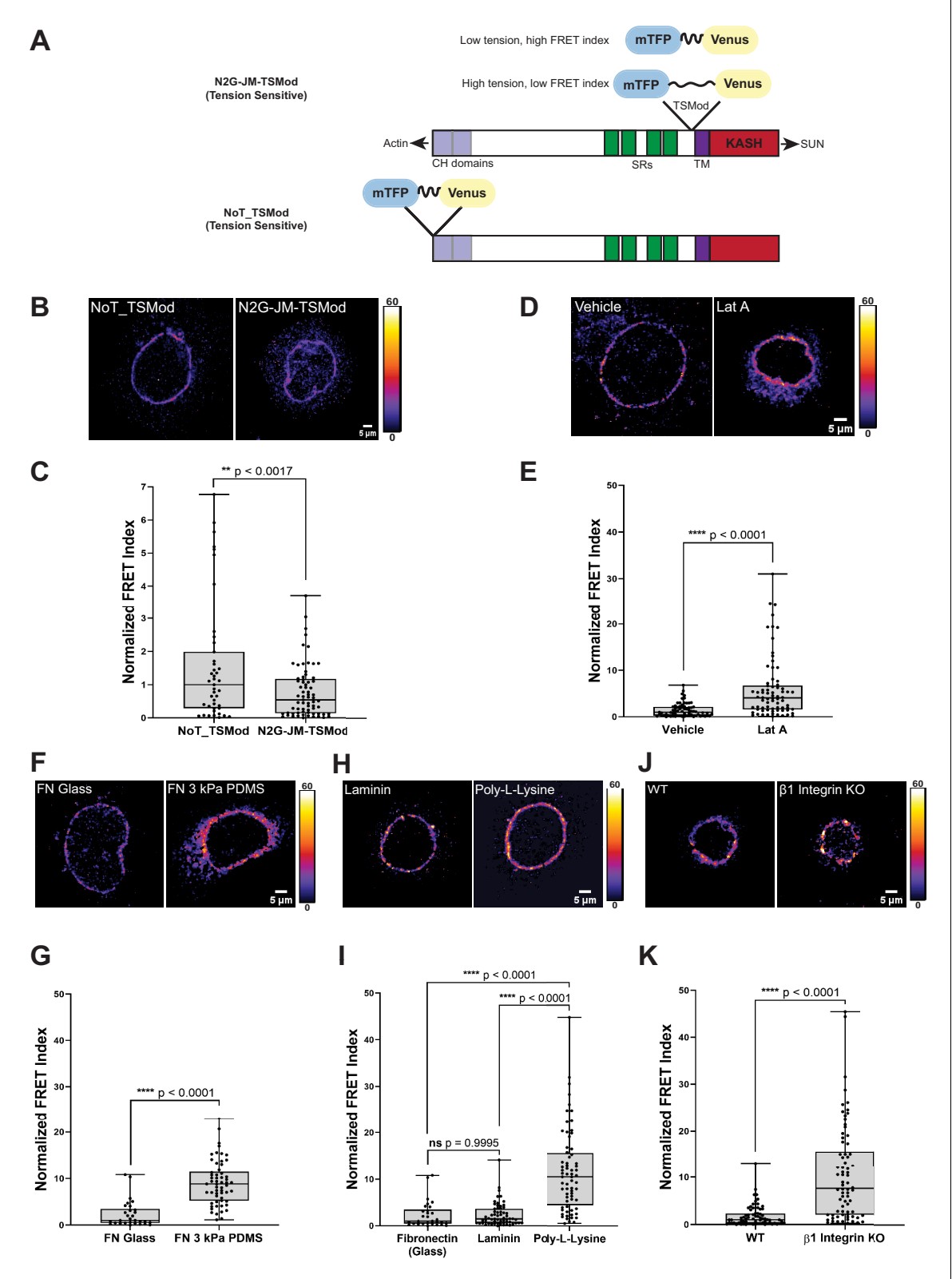

**Figure 1.** A tension sensor in mini-Nesprin-2 is under actin-dependent tension when mouse keratinocytes (MKCs) engage the extracellular matrix through integrins. (**A**) Cartoon of the N2G-JM-TSMod, in which the mTFP-Venus tension sensor module is inserted between the cytoplasmic domain and the C-terminal KASH domain/transmembrane domain of mini-Nesprin-2. Tension leads to a reduction in the fluorescence resonance energy transfer (FRET) index. Cartoon of the NoT_TSMod control, in which the TSMod resides at the N-terminus of the N2G, and therefore cannot experience

*Figure 1 continued on next page*

*Figure 1 continued*

intramolecular tension. (B, C) The N2G-JM-TSMod displays higher tension (lower FRET index) than the no-tension NoT_TSMod control (higher FRET index) when expressed in MKCs plated on FN-coated glass. Representative images show that, at low expression levels, both the N2G-JM-TSMod and NoT_TSMod are successfully targeted to the nuclear envelope. Images are pseudocolored according to the normalized FRET index (*Feige et al., 2005*). The median FRET index value of cells expressing the NoT_TSMod was set to a value of 1 in (C) and used to scale to relative values for the N2G-JM-TSMod. (D, E) Disruption of actin filaments with 0.5 μM latrunculin A (Lat A) decreases the tension on the N2G-JM-TSMod, leading to a higher FRET index. The median FRET index of cells expressing the N2G-JM-TSMod was set to a value of 1 in (E). (F, G) Tension on the N2G-JM-TSMod is sensitive to substrate mechanics. Plating of MKCs on FN-coated compliant substrates (3 kPa PDMS) leads to increased FRET compared to MKCs plated on FN-coated glass. (H, I) Plating of MKCs on extracellular matrix that engages integrins drives high tension on the N2G-JM-TSMod. MKCs grown on glass coated with fibronectin or laminin drive a higher tension state (low FRET index) than for cells grown on glass coated with poly-L-lysine (high FRET index). The median FRET index value of cells expressing the N2G-JM-TSMod plated on FN-coated glass was set to a value of 1 in (G) and (I). Data for FN replotted from (I). (J, K) MKCs lacking β1 integrin fail to drive high tension on the N2G-JM-TSMod. Representative images demonstrate that the N2G-JM-TSMod is successfully targeted to the nuclear envelope in wild-type (WT) and β1 integrin null MKCs. Higher FRET indexes at the nuclear envelope in β1 integrin null MKCs demonstrate that β1 integrin engagement with the extracellular matrix is required for high tension on the N2G-JM-TSMod. The median FRET index value of cells expressing the N2G-JM-TSMod plated on fibronectin-coated glass was set to a value of 1 in (K). For all plots, errors reflect SD, $n \geq 30$ cells for each condition measured from $n = 3$ experiments. ****p<0.0001 as determined by unpaired t-test (C, E, G, K) or one-way ANOVA (I). All scale bars = 5 μm.

The online version of this article includes the following figure supplement(s) for figure 1:

**Figure supplement 1.** Validation controls for the N2G-JM-TSMod.

index (*Figure 1A*). Indeed, relative to the mean NoT_TSMod FRET ratio (set to a value of 1), the N2G-JM-TSMod exhibited lower FRET indexes when expressed in undifferentiated MKCs as assessed qualitatively by the pseudocolored images reflecting the relative FRET index (*Figure 1B*) or quantitatively across populations of MKCs (*Figure 1C*). To determine if the actin cytoskeleton is required for tension on the N2G-JM-TSMod in undifferentiated MKCs, we pharmacologically disrupted filamentous actin with latrunculin A (Lat A). After 5 hr of Lat A treatment, the N2G-JM-TSMod displayed relaxed tension compared to the vehicle control (set to a value of 1) as indicated by an increase in the FRET indexes (*Figure 1D*, E). By contrast, Lat A treatment had no effect on the NoT_TSMod, as expected (*Figure 1—figure supplement 1B*). Based on these data, we conclude that (1) the tension reported by the N2G-JM-TSMod reflects intramolecular tension exerted on the juxtamembrane region and (2) the N2G-JM-TSMod is sensitive to actin-dependent tension at the nuclear envelope.

Next, we examined how the properties of cell-substrate engagement influenced tension on the LINC complex. We plated undifferentiated MKCs on glass or 3 kPa polydimethylsiloxane (PDMS) substrates coated with fibronectin (FN) (*Mertz et al., 2013*). We observed high tension on the N2G-JM-TSMod on stiff glass substrates (set to a value of 1) relative to cells plated on the 3 kPa substrates, which displayed lower tension (higher FRET ratio; *Figure 1F*, G). Proliferative keratinocyte progenitor cells bind to the basement membrane via several integrin-based adhesions with the laminin receptors α3β1 and the hemidesmosome-specific α6β4 being the chief epidermal integrins (reviewed in *Burgeson and Christiano, 1997*). Thus, we also explored how substrate composition affects tension on the LINC complex by plating MKCs on either FN- or laminin-coated glass coverslips or those coated with poly-L-lysine, which allows cell adhesion through its positive charge rather than specific cellular adhesions. MKCs expressing the N2G-JM-TSMod exhibited high tension (lower relative FRET ratio) at the nuclear envelope when plated on fibronectin or laminin relative to when MKCs were plated on poly-L-lysine (*Figure 1H*, I). Furthermore, *β1* integrin (*Itgb1*) null MKCs plated on glass coverslips coated with FN had higher N2G-JM-TSMod FRET ratios (lower tension) than WT controls (*Figure 1J*, K). As MKCs lacking *β1* integrin expression possess high levels of actomyosin contractility and can still engage with the substrate through *β6* integrins (*Bandyopadhyay et al., 2012*; *Raghavan et al., 2003*), these observations suggest that *β1* integrin engagement is explicitly required for high tension on LINC complexes at the nuclear envelope in epidermal progenitor cells.

## Cell-intrinsic integrin engagement predicts N2G-JM-TSMod tension and differentiation in cohesive MKC colonies

Primary MKCs can be induced to differentiate in vitro by elevating extracellular calcium levels, leading to the formation of cohesive colonies that engage E-cadherin-based cell-cell adhesions and

subsequently differentiate as assessed by upregulation of markers expressed in epidermal supra-basal layers in vivo (*Mertz et al., 2013*). In our previous work, we demonstrated that this transition leads to a reorganization of focal adhesions and traction stresses to cells at the colony periphery, while cells in the colony interior solely engage cell-cell adhesions (*Mertz et al., 2013*; *Figure 2A*). We also documented that cells at the colony periphery display a biased nuclear position toward the colony center while cells within the colony interior display a central nuclear position (*Figure 2A*), suggesting that LINC complex tension might be distinct between MKCs at the colony periphery and interior (*Stewart et al., 2015*). To test if tension on the N2G-JM-TSMod is sensitive to the presence of cell-intrinsic focal adhesion engagement, we segmented cells in cohesive MKC colonies into 'periphery' and 'interior' cells. We find that the FRET ratio of the N2G-JM-TSMod for interior cells is significantly higher than that observed for cells at the colony periphery (*Figure 2B*), indicating that tension on the LINC complex is lost in interior cells that contain solely cell-cell adhesions. These data are consistent with the ability of focal adhesions to drive LINC complex tension in a cell-intrinsic manner in undifferentiated MKCs and reinforce that actomyosin contractility and stress fibers, which are present in both periphery and interior cells, are not sufficient to induce a high-tension state on LINC complexes in contexts where integrins are not engaged.

Given that the LINC complex has been shown to transmit tension from the cell substrate to the lamin network at the nuclear periphery (*Ihalainen et al., 2015*), we examined whether tension on lamin A/C within cohesive MKC colonies mirrored that observed for the LINC complex with the N2G-JM-TSMod. To this end, we employed a conformationally sensitive lamin A/C antibody, which has been used to examine changes in lamin tension due to substrate stiffness (*Ihalainen et al., 2015*). When A-type lamins are under tension, the epitope of lamin A/C becomes inaccessible, resulting in a decrease in fluorescence. We immunostained differentiated, cohesive MKC colonies with a conformationally sensitive lamin A/C antibody and examined cells at the periphery and interior. We imaged nuclei in which we could resolve the apical and basal regions of the nuclear envelope and quantified the fluorescent signal in the z-plane as described (*Ihalainen et al., 2015*; *Figure 2C*). We fitted the signal with two gaussian curves and extracted the ratio of the basal to apical intensity across many colonies (n = 17). In cells at the colony periphery, we observe a strong bias of fluorescence at the apical nuclear surface compared to the basal surface (*Figure 2D*), indicating that tension on the lamin network is high on the basal surface in cells at the periphery of colonies. By contrast, cells in the colony interior display far more uniform conformationally sensitive lamin A/C staining (*Figure 2E*). Taken together, these results suggest that cells at the colony periphery that engage focal adhesions have high tension on both LINC complexes and the nuclear lamina, while cells at the colony interior possess relaxed LINC complexes and lamin A/C.

To examine whether the cells in the colony interior are more likely to express differentiation markers, we performed RNA fluorescence in situ hybridization (FISH) for *Sprr1b* mRNA, which is upregulated in differentiated MKCs and is found in the EDC. We find that MKCs at the colony interior express more *Sprr1b* and *Ivl* (involucrin) mRNA 24 and 48 hr after induction of differentiation compared to cells at the colony periphery (*Figure 2F, G*, *Figure 2—figure supplement 1*). These data indicate that tension on LINC complexes and the nuclear lamina is more prominent in cells that maintain integrin adhesions and a progenitor fate.

## The epidermis of Sun dKO mice displays precocious differentiation despite normal adhesion

To determine whether tension on LINC complexes controls MKC differentiation, we analyzed the skin of mouse models lacking the ubiquitously expressed SUN proteins, SUN1 and SUN2, which are expected to lack all LINC complexes in somatic tissues (*Zhang et al., 2009*; *Zhang et al., 2007*). Similar to our prior analysis of postnatal mouse skin (*Stewart et al., 2015*), SUN1 and SUN2 are expressed at the nuclear envelope in all cellular layers of the E15.5 developing epidermis (*Figure 3—figure supplement 1A*). Although *Sun1^{-/-}/Sun2^{-/-}* (hereafter Sun dKO) mice die after birth (*Zhang et al., 2009*), we were able to analyze the developing epidermis of Sun dKO mice at E15.5. Western blot analysis revealed that SUN1 and SUN2 proteins were absent in the skin of Sun dKO compared to wild-type (WT mice that express both SUN proteins (*Figure 3—figure supplement 1B*)).

Histological analysis of E15.5 skin tissue revealed a thickening of the epidermis in the Sun dKO mouse compared to WT littermates (*Figure 3A*). Immunostaining of skin sections of WT and Sun

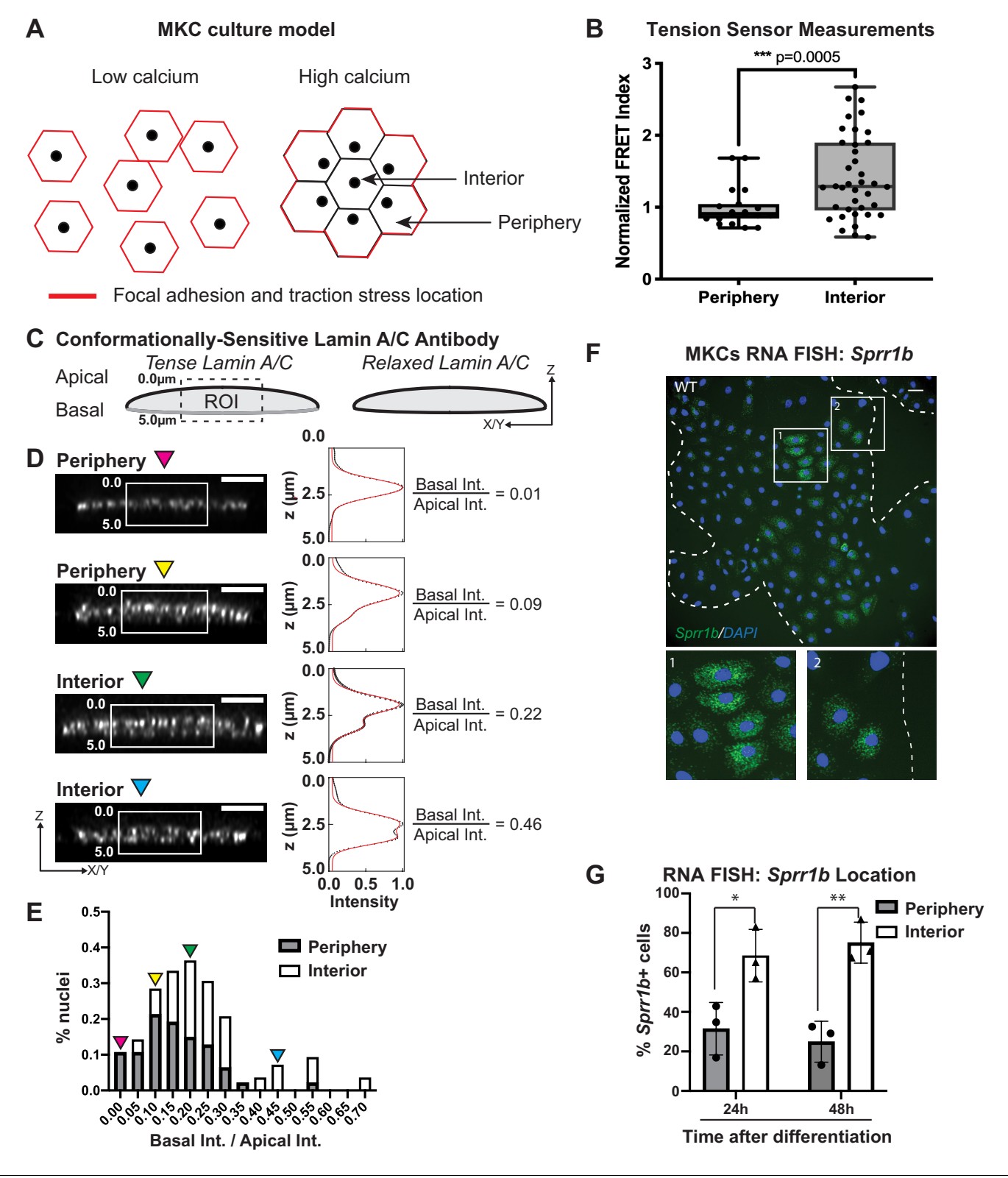

**Figure 2.** Tension on the N2G-JM-TSMod and nuclear lamina is released at the interior of cohesive mouse keratinocyte (MKC) colonies concomitant with differentiation. (A) Cartoon of cell junction reorganization upon differentiation. In single MKCs grown in low calcium media, focal adhesions and traction forces are found at the periphery of each cell (red). In response to high calcium, MKCs form cohesive colonies and engage cell-cell junctions, which leads to a reorganization of focal adhesions and traction forces to the colony periphery. Cells in the colony interior have cell-cell but not cell-

*Figure 2 continued on next page*

*Figure 2 continued*

matrix adhesions. (B) The fluorescence resonance energy transfer index is higher in cells at the interior of cohesive MKC colonies compared to cells at the periphery, suggesting that tension on the N2G-JM-TSMod requires cell-intrinsic focal adhesion engagement. (C) Cartoon of method used to measure changes in the tension state of A-type lamins using a conformationally sensitive lamin A/C antibody whose epitope is lost when lamins are under tension. Cells under tension (left) lose staining on the basal side of the nuclear envelope. The profile of fluorescence intensity from the apical to basal side of each nuclei was measured for a region of interest in XZ and YZ slices from individual nuclei at varying locations within a colony. (D) Tension at the basal surface of nuclei is lost at the interior of cohesive MKC colonies and is maintained in cells at the colony periphery in response to differentiation. Examples of XZ and YZ confocal sections of individual nuclei and corresponding Z-intensity profiles measured in the region of interest, indicated by the white box (left). Intensity profiles were fit to two gaussians (right), representing staining of the apical and basal sides of the nuclear envelope. The intensity of antibody staining was defined as the area under the curve of the gaussian distribution corresponding to each side of the nuclear envelope. A ratio of the intensity of antibody staining of the basal relative to the apical side of the nuclear envelope was calculated. This analysis revealed that tension on the basal nuclear surface is relaxed at the colony interior relative to the colony periphery. Scale bar = 5 µm. (E) A histogram of the ratio of basal to apical intensity calculated as described in (C) for all cells analyzed (n = 47 for periphery and n = 28 for interior) shows that the ratio of basal to apical intensity for interior cells is shifted to higher values relative to cells at the periphery, indicating the lamina is under less tension. Values were binned every 0.05 arbitrary units, and the central value of each bin is labeled. Representative images for the low (magenta triangle) and high (blue triangle) bins, and bins corresponding to the highest percentage of periphery (yellow triangle) and interior (green triangle) are shown in (C). (F, G) The differentiation marker *Sprr1b* is expressed at higher levels in the colony interior than at the colony periphery. (F) Representative image of RNA fluorescence in situ hybridization for *Sprr1b* 24 hr after addition of calcium to induce differentiation. Inset 1 shows *Sprr1b*-positive cells at the colony interior. Inset 2 shows *Sprr1b*-positive and -negative cells at the colony periphery. Dotted lines are colony outline. Scale bar = 100 µm. (G) Quantitation of the percent of *Sprr1b*-positive cells that are located at the interior and periphery of WT MKC colonies normalized to the total *Sprr1b*-positive cells. 24h: 24 hr calcium treatment; 48h: 48 hr calcium treatment. *p<0.05. **p<0.01 as determined by unpaired t-test. Error bars are SD. N = 3 biological replicates.

The online version of this article includes the following figure supplement(s) for figure 2:

**Figure supplement 1.** The differentiation marker involucrin (*Ivl*) is expressed at higher levels in the colony interior than at the colony periphery.

dKO mice with antibodies against proteins expressed in differentiated keratinocytes revealed an expansion of keratin 10 (K10) and K1 in the spinous layer, involucrin in the spinous and granular layers, and fillagrin in the granular and cornified layers (*Figure 3B*, C, *Figure 3—figure supplement 2*). These observations suggest that LINC complex ablation in the epidermis could enhance keratinocyte differentiation.

To further examine whether the thickened epidermis of Sun dKO mice resulted from changes in differentiation and/or proliferation, we performed an EdU pulse chase experiment. We labeled proliferating cells in embryonic WT and Sun dKO mice with EdU and examined the skin after 24 hr (*Figure 3D*). Immunostaining skin sections with K10 antibodies and staining for EdU incorporation revealed that the total number of EdU-positive cells was similar between WT and Sun dKO epidermis, indicating that proliferation was not altered in the absence of LINC complexes in skin (*Figure 3E*). However, while 40% of EdU-positive cells had moved to the suprabasal, K10-positive layer in WT tissue (*Figure 3F*), 60% of EdU-positive cells were observed in the differentiated layers in Sun dKO skin (*Figure 3D, F*), indicating an increase in keratinocyte differentiation in the absence of LINC complexes. Thus, Sun dKO keratinocytes precociously differentiate in vivo without changes in proliferation.

As integrin signals are required to repress the differentiation of basal keratinocytes (*Adams and Watt, 1989*; *Levy et al., 2000*; *Zhu et al., 1999*), we explored the possibility that cell-matrix adhesions are altered in number and/or size in the epidermis of Sun dKO mice. We therefore performed electron microscopy on skin sections from heterozygous (*Sun1/2+/-*) and Sun dKO mice at P0.5. The basal layer-basal lamina interface in *Sun1/2+/-* and Sun dKO skin appeared indistinguishable, displaying tight association in mice from both genotypes (*Figure 3G*). Moreover, hemidesmosomes (which can be observed in electron micrographs, arrows) were of equivalent number and size (*Figure 3H*). These observations suggest that Sun dKO mice display precocious epidermal differentiation in vivo that is uncoupled from changes in adhesion between basal keratinocytes and the basal lamina.

## Sun dKO MKCs exhibit precocious differentiation in vitro

To further analyze keratinocyte differentiation, we isolated WT and Sun dKO MKCs from newborn mice and performed global transcriptional profiling on WT and Sun dKO MKCs grown in either low calcium media to maintain a progenitor state or high calcium media to induce differentiation. Gene

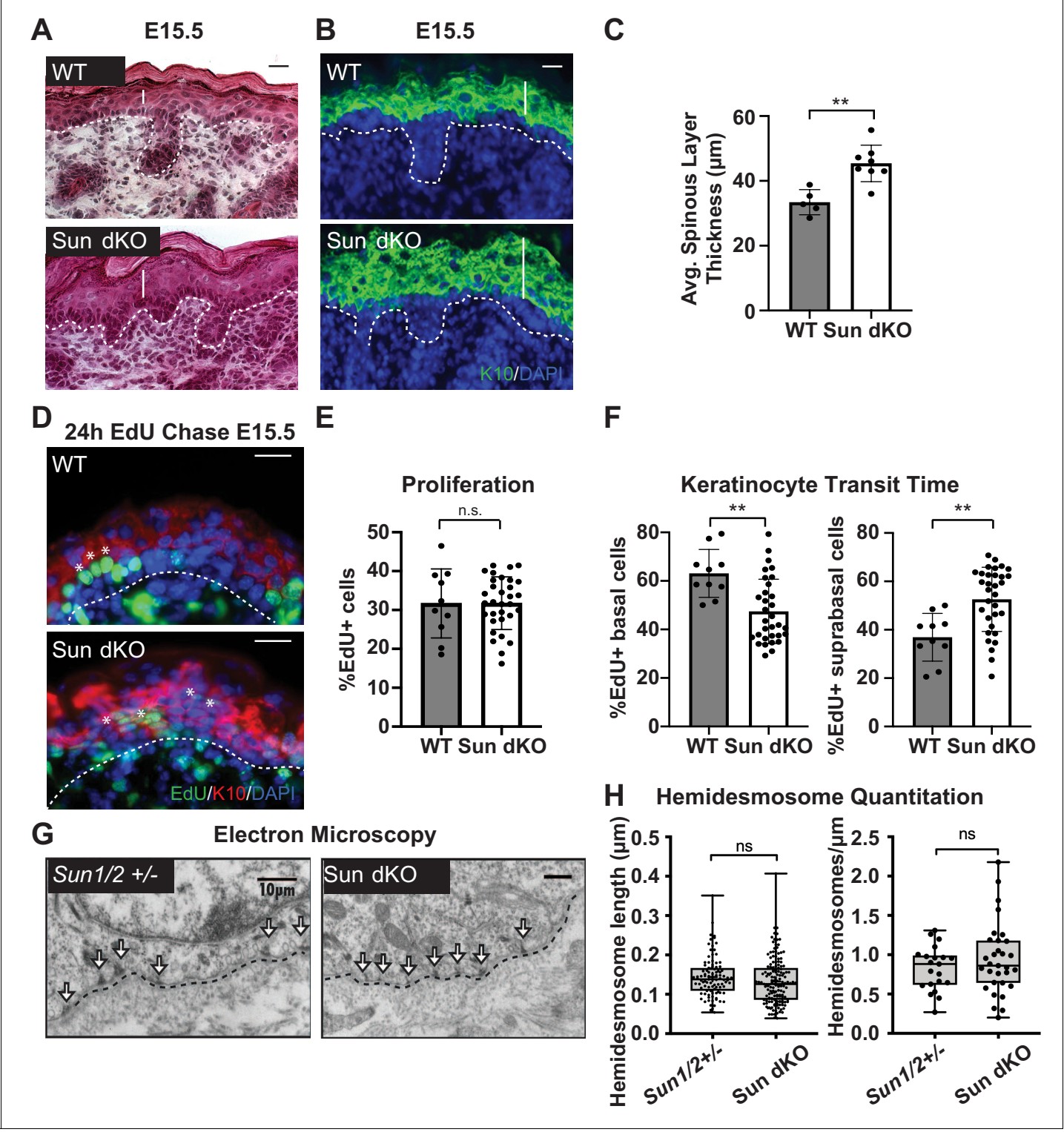

**Figure 3.** Linker of nucleoskeleton and cytoskeleton complex ablation leads to precocious epidermal differentiation in vivo. (**A**) Increased epidermal thickness in Sun dKO mice. Representative hematoxylin and eosin staining of WT and Sun dKO skin at age E15.5. Dotted line denotes dermal/epidermal junction. Vertical line denotes epidermal thickness. Scale bar = 20 µm. (**B**) Expansion of the spinous layer in Sun dKO mice. Representative immunostaining for keratin 10 (K10) in WT and Sun dKO skin at age E15.5. Dotted line denotes dermal/epidermal junction. Vertical line denotes spinous layer thickness. Scale bar = 20 µM. Nuclei are stained with DAPI. (**C**) Quantitation of average spinous layer thickness as determined by immunostaining for K10 in WT and Sun dKO epidermis at E15.5. Unpaired t-test was used to determine statistical significance. **p<0.01. Error bars are

*Figure 3 continued on next page*

*Figure 3 continued*

SD. N = 5–8 biological replicates per genotype. (D–F) Pulse chase analysis reveals an increase in EdU-positive cells in the suprabasal layers of Sun dKO mice (and a decrease in EdU-positive basal cells) but no increase in overall proliferation. (D) Representative images of E15.5 WT and Sun dKO skin 24 hr after EdU pulse. Spinous layer keratinocytes are marked by K10 staining. Dotted line denotes dermal/epidermal junction. Asterisks denote suprabasal keratinocytes marked with EdU. Scale bar = 20 µm. Nuclei are stained with DAPI. (E) Quantitation of total EdU-positive cells normalized to total epidermal cells in both WT and Sun dKO epidermis at E15.5. Statistical significance was determined using unpaired t-test. ns: not significant. N = 3 mice/genotype. (F) Quantitation of location of EdU-positive cells in the epidermis of E15.5 WT and Sun dKO epidermis 24 hr after EdU pulse. K10 staining was used to differentiate suprabasal from basal keratinocytes. Statistical significance was determined by performing unpaired t-test. **p<0.01. N = 3 mice/genotype. (G, H) Adhesion between basal keratinocytes and the basal lamina is normal in the Sun dKO skin. (G) Representative electron microscopy images of P0.5 WT and Sun dKO basal keratinocytes in vivo. Scale bar = 10 µm. Dotted line denotes basement membrane. Arrows point to hemidesmosomes. (H) Quantitation of hemidesmosome length (left) and the number of hemidesmosomes per surface area (right) in control and Sun dKO epidermis at age P0.5. ns: not statistically significant as determined by unpaired t-test.

The online version of this article includes the following figure supplement(s) for figure 3:

**Figure supplement 1.** SUN1 and SUN2 are expressed throughout the epidermis.
**Figure supplement 2.** Thickening of the differentiated layers in the Sun dKO epidermis.

ontology analysis revealed striking differences in the expression of genes associated with keratinocyte differentiation between WT and Sun dKO MKCs in both low and high calcium, including those that reside in the EDC (*Figure 4A*, B and *Supplementary files 1* and *2*). Further examination of these RNAseq data revealed evidence of precocious differentiation in Sun dKO MKCs. For example, numerous members of the gene family in the SPRR region of the EDC (*Figure 4A*) were de-repressed in Sun dKO MKCs grown in the low calcium condition (*Figure 4C*). We validated this global analysis by RT-qPCR analysis of *Sprr* genes (*Figure 4D*, *Figure 4—figure supplement 1A*). Not only do we observe precocious expression of genes such as *Sprr1b* in Sun dKO MKCs cultured in low calcium media, but we also observe much higher expression in the presence of calcium in Sun dKO MKCs (*Figure 4D*, *Figure 4—figure supplement 1A*). RNA FISH analysis of the level of *Sprr1b* (*Figure 4E*, *Figure 4F*) and *Ivl* (*Figure 4—figure supplement 1B, C*) transcripts further confirmed that Sun dKO MKCs displayed higher expression levels of differentiation genes compared to WT MKCs.

Since we found that the differentiated, interior MKCs within cohesive colonies displayed low tension on the LINC complex and A-type lamins (*Figure 2*), we hypothesized that the lack of LINC complexes may result in differentiation of progenitor cells at the colony periphery despite integrin engagement, which is normally associated with maintenance of the progenitor fate. To address this hypothesis, we performed RNA FISH on WT and Sun dKO cohesive MKC colonies. While we demonstrated that WT MKCs biased expression of differentiation markers such as *Sprr1b* and *Ivl* at the colony interior over the colony periphery (*Figure 2F*, G) in agreement with repressive signals from integrin engagement in cells at the colony periphery, Sun dKO MKC colonies instead expressed *Sprr1b* mRNAs in MKCs residing in both the interior and periphery (*Figure 4E*), despite still having focal adhesions in peripheral cells (*Figure 4—figure supplement 2*). As a consequence, expression of differentiation markers is both elevated at the colony periphery and is random with respect to colony position in the absence of LINC complexes (*Figure 4H*, *Figure 4G*). Finally, we directly tested if β1 integrins are required for proper keratinocyte differentiation by performing RT-PCR to analyze *Sprr1b*, Involucrin (*Ivl*), and *S100a14* mRNA levels in β1 integrin null MKCs. β1 integrin null MKCs displayed precocious expression of all three of these epidermal differentiation genes (*Figure 4I*), mirroring our observations in Sun dKO MKCs. Taken together, these results suggest that LINC complexes are required to repress epidermal differentiation genes when integrins are actively engaged.

## Sun dKO MKCs exhibit an increase in chromatin accessibility at genes within the EDC

The EDC moves away from the nuclear lamina during epidermal differentiation (*Gdula et al., 2013*; *Mardaryev et al., 2014*; *Williams et al., 2002*) with an associated loss of repressive H3K27me3 chromatin marks at differentiation-specific genes (*Lien et al., 2011*). Given that Sun dKO MKCs display a precocious differentiation phenotype, we hypothesized that the EDC may be more accessible when LINC complexes are ablated. To test this, we applied assay for transposase-accessible

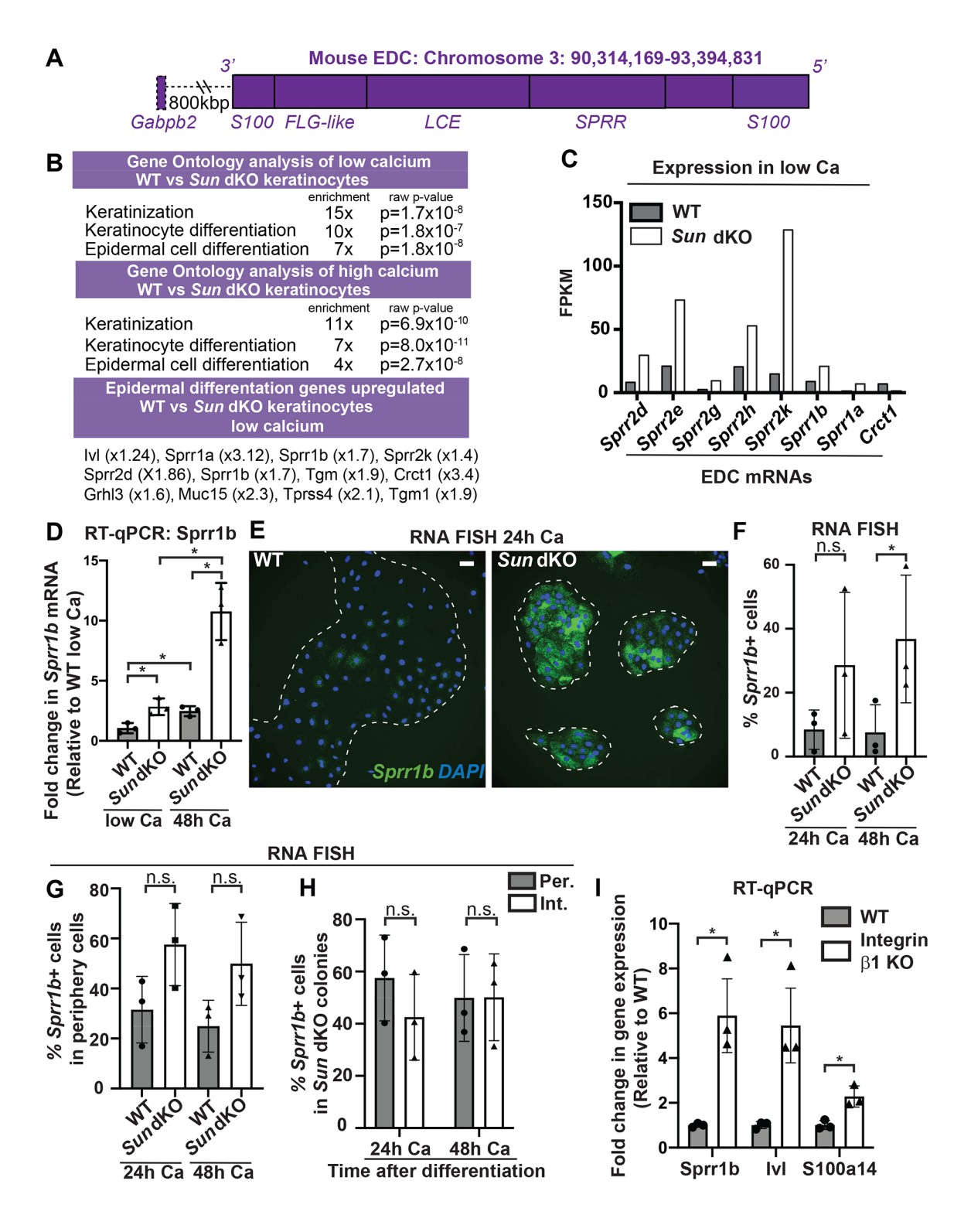

**Figure 4.** Linker of nucleoskeleton and cytoskeleton complex ablation leads to precocious mouse keratinocyte (MKC) differentiation in vivo, which is associated with aberrant upregulation of differentiation markers at the colony periphery. (**A**) Cartoon of the epidermal differentiation complex (EDC), a large genic region that is coordinately upregulated upon epidermal differentiation. (**B**) Comparative transcriptome analysis reveals that Sun dKO MKCs display precocious expression of epidermal differentiation genes when cultured in low calcium media and higher levels of expression of epidermal

*Figure 4 continued on next page*

*Figure 4 continued*

differentiation genes in high calcium media. See also *Supplementary files 1* and *2*. (C) Examples of EDC genes that are precociously expressed in Sun dKO MKCs cultured in low calcium media from the RNAseq data, expressed as fragments per kb of transcript per million reads. (D) Real-time qPCR analysis of *Sprr1b* in WT and Sun dKO MKCs in the presence and absence of calcium validates precocious *Sprr1b* expression in Sun dKO MKCs without calcium stimulation. Ct values were normalized to GAPDH. Fold change in expression was determined by calculating the $2^{\Delta\Delta Ct}$ relative to the mean $\Delta Ct$ of WT MKCs cultured in low calcium media. Statistical significance was determined by performing multiple t-tests. * p<0.05. The Holm–Sidak method was used to correct for multiple comparisons. Error bars are SD. N = 3 biological replicates. (E–G) Cohesive Sun dKO MKCs express elevated levels of differentiation markers and lose the relationship between position in the colony and EDC gene expression. (E) Representative images of RNA fluorescence in situ hybridization (FISH) for *Sprr1b* in WT and Sun dKO MKCs after 24 hr calcium treatment. Dotted lines are colony outline. Scale bar = 100 μm. (F) Quantitation of RNA FISH for *Sprr1b* in WT and Sun dKO MKCs after calcium treatment. *Sprr1b*-positive cells were counted and normalized to total cells in each field. 24h: 24 hr calcium treatment; 48h: 48 hr calcium treatment. Statistical significance was determined by performing unpaired t-test. *p<0.05. ns: not statistically significant. Error bars are SD. N = 3 biological replicates. (G) Quantitation of the percent of *Sprr1b*-positive cells that are located at the periphery of WT and Sun dKO MKC colonies normalized to the total *Sprr1b*-positive cells. 24 hr Ca: 24 hr calcium treatment; 48 hr Ca: 48 hr calcium treatment; ns: not statistically significant as determined by unpaired t-test. Error bars are SD. N = 3 biological replicates. (H) Quantitation of the percent of *Sprr1b*-positive cells that are located at the interior and periphery of Sun dKO MKC colonies normalized to the total *Sprr1b*-positive cells. 24h: 24 hr calcium treatment; 48h: 48 hr calcium treatment; ns: not statistically significant as determined by unpaired t-test. Error bars are SD. N = 3 biological replicates. (I) Real-time qPCR analysis of *Sprr1b*, involucrin (*Ivl*), and *S100a14* demonstrate precocious differentiation in β1 integrin null (KO) MKCs. Ct values were normalized to GAPDH. Fold change in expression was determined by calculating the $2^{\Delta\Delta Ct}$ relative to the mean of WT $\Delta Ct$. Statistical significance was determined by Student's t-tests. *p<0.05. Error bars are SD. N = 3 biological replicates. The online version of this article includes the following figure supplement(s) for figure 4:

**Figure supplement 1.** Additional evidence for precocious differentiation in Sun dKO mouse keratinocytes (MKCs).
**Figure supplement 2.** Cohesive Sun dKO mouse keratinocyte (MKC) colonies have focal adhesions at the colony periphery.

chromatin using sequencing (ATAC-seq) (*Figure 5—figure supplement 1* and *Supplementary file 3*) to determine if chromatin accessibility was changed in Sun dKO MKCs with a focus on the EDC. Globally, most genes with annotated ATAC-seq peak(s) are conserved between WT and Sun dKO MKCs cultured in low calcium media (9307 genes, *Figure 5—figure supplement 1* and *Supplementary file 4*). Sun dKO MKCs showed a loss of 1070 genes with annotated ATAC-seq peaks compared to WT MKCs, which are outnumbered by gains in Sun dKO MKCs (2935 genes; *Figure 5—figure supplement 1* and *Supplementary file 4*). Taken together, we conclude that there is a relatively modest increase in global chromatin accessibility in Sun dKO MKCs.

It is therefore striking that Sun dKO MKCs displayed almost twice as many ATAC-seq peaks within the EDC region (34 peaks) relative to WT MKCs (19 peaks) (*Figure 5*). Specifically, in Sun dKO MKCs we observed 18 additional peaks, 16 maintained peaks, and a loss of just 3 peaks compared to WT MKCs (*Figure 5A*). This net gain of peaks is observed throughout all regions of the EDC, with the SPRR region displaying the most striking change, gaining five novel ATAC-seq peaks upon loss of SUN proteins (*Figure 5B*, C). Importantly, while there is a trend toward increased chromatin accessibility in regions flanking the EDC (*Figure 5—figure supplement 2A*) in line with a trend toward increased accessibility across the genome (*Figure 5—figure supplement 1*), the gain of ATAC-seq peaks is more pronounced in the EDC compared to chromatin regions 5′ and 3′ (*Figure 5D*). We also noted examples in which the additional ATAC-seq peak was located near the promoter region of epidermal differentiation genes such as *Sprr2a3* (*Figure 5—figure supplement 2B*). We next investigated if there are changes in chromatin accessibility at other genes tied to epidermal differentiation that reside outside the EDC. We observed a trend toward additional ATAC-seq peaks in some genic regions encoding keratins and cell adhesion genes tied to epidermal differentiation in Sun dKO MKCs (*Supplementary file 5*). We further asked if Sun dKO MKCs displayed hallmarks of repression of cell proliferation genes, which might be expected upon loss of MKC progenitors due to precocious keratinocyte differentiation. However, no such trend was observed (*Supplementary file 5*). Thus, the changes in chromatin accessibility support a model in which keratinocyte differentiation genes are precociously expressed in Sun dKO cells despite largely normal accessibility at genes that underlie progenitor maintenance.

To test if this mode of regulation is specific to keratinocyte differentiation, we also looked at the chromatin state upon loss of SUN proteins at other master transcriptional regulators that have previously been shown to be repressed in basal keratinocytes (*Ezhkova et al., 2009*). Indeed, chromatin at the master transcriptional regulators for muscle (*Myod1*) and the HOX (*Hoxd10*, *Hoxb13*, and *Hoxa11*) gene families is in a closed chromatin state in both WT and Sun dKO MKCs (*Figure 5—*

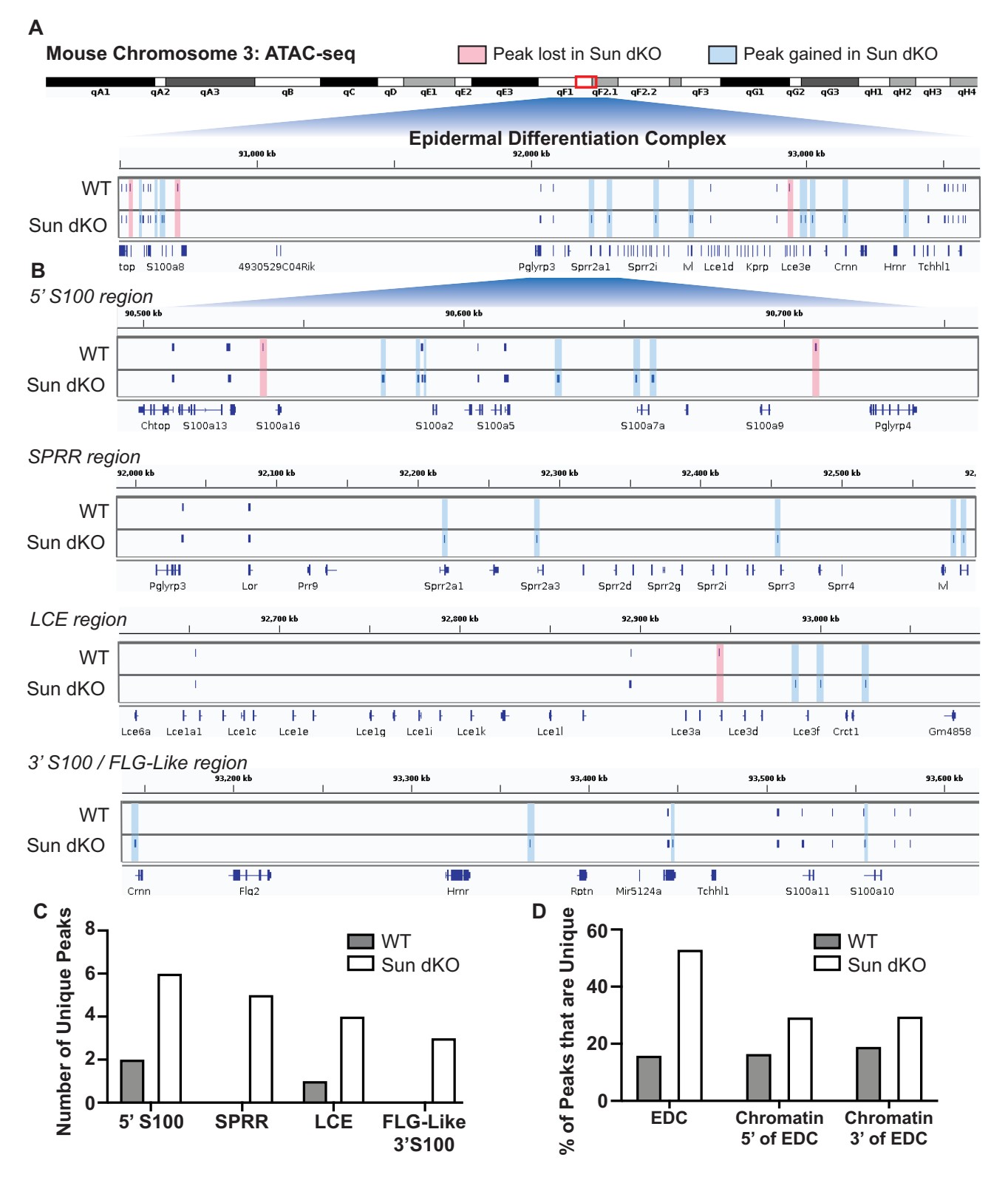

**Figure 5.** Linker of nucleoskeleton and cytoskeleton complex ablation leads to an increase in chromatin accessibility of genes residing in the epidermal differentiation complex (EDC). Assay for transposase-accessible chromatin using sequencing (ATAC-seq) tracks generated from either WT (top; N = 2) or Sun dKO (bottom; N = 2) mouse keratinocytes (MKCs) grown in low calcium media. (**A**) There is an increase in the number of accessible peaks (n = 38) within the EDC in Sun dKO MKCs compared to WT MKCs (n = 19). Peaks gained in Sun dKO MKCs are highlighted in blue, and peaks lost are

*Figure 5 continued on next page*

*Figure 5 continued*

highlighted in red. Unchanged peaks are not highlighted. (B) Expanded view of ATAC-seq peaks in each indicated region of the mouse EDC. Peaks gained in Sun dKO MKCs are highlighted in blue, and peaks lost are highlighted in red. Unchanged peaks are not highlighted. (C) Quantitation of novel ATAC-seq peaks in WT and Sun dKO MKCs. The number of unique peaks over the indicated regions is plotted for the two genotypes. (D) Quantitation of the percentage of total ATAC-seq peaks that are novel in WT and Sun dKO MKCs for the EDC and surrounding chromatin regions (ATAC-seq tracks in *Figure 5—figure supplement 1*). While a trend toward gains in ATAC-seq peaks is observed in Sun dKO MKCs, the gain of novel peaks at the EDC is more pronounced than in equivalently sized chromatin regions 5′ and 3′ to the EDC. See also *Supplementary file 3*. The online version of this article includes the following figure supplement(s) for figure 5:

**Figure supplement 1.** Assay for transposase-accessible chromatin using sequencing (ATAC-seq) analysis of WT and Sun dKO mouse keratinocytes (MKCs).

**Figure supplement 2.** Increased accessibility within the epidermal differentiation complex (EDC) is specific as assessed by the assay for transposase-accessible chromatin using sequencing (ATAC-seq) in Sun dKO mouse keratinocytes (MKCs) cultured in low calcium media.

figure supplement 2C). Two of four neuronal master transcription factors (*Olig2* and *Neurog2*) also remained in a closed state while the others (*Olig3* and *Neurog2*) revealed a novel peak in Sun dKO cells (*Figure 5—figure supplement 1*). Taken together, our observations suggest that LINC complexes are required to maintain a closed, inhibited state of the EDC in undifferentiated keratinocytes, providing a mechanism by which loss of LINC complexes leads to precocious keratinocyte differentiation.

## Discussion

Here, we demonstrate that LINC complex tension in MKCs responds specifically to β1 integrin engagement in a cell-intrinsic manner. While β1 integrin-dependent signals normally repress the differentiation of keratinocytes, in the absence of functional LINC complexes these basal progenitors instead differentiate precociously both in vitro and in vivo. Based on our findings, we propose a model in which LINC complexes transmit forces from the β1 integrin-engaged actin network to the nuclear lamina to directly regulate the expression of epidermal differentiation genes (*Figure 6*), although an indirect role for the LINC complex in an alternative signaling pathway remains possible. In particular, as our electron microscopy data strongly suggest that adhesion between basal keratinocytes and the basal lamina remains normal in Sun dKO animals, we strongly favor this direct model (*Figure 6*). Of note, cells migrating through constrictions or plated at high versus low packing density display lower tension on a distinct Nesprin TSMod variant similar in design to the one described here, suggesting that tension on the LINC complex may respond to a number of mechanical inputs depending on the experimental system (*Déjardin et al., 2020*). However, high tension on the LINC complex at the edge of cell monolayer upon wounding compared to the interior of the monolayer (*Déjardin et al., 2020*) is in line with the integrin dependence observed in this study.

Interestingly, our data indicate that high Rho activity and actin contractility are not sufficient to drive tension on the LINC complex – indeed, this represents a critical point that lies at the heart of our model. For instance, cells at the interior of cohesive MKC colonies also display extensive stress fibers integrated at cell-cell adhesions (*Mertz et al., 2013*; *Stewart et al., 2015*), but nonetheless show low tension on the LINC complex and the nuclear lamina, as demonstrated in *Figure 2A–D*. These observations suggest that actomyosin contractility is not sufficient, in and of itself, to drive LINC complex tension. This point is underscored by our finding that the LINC complex is relaxed in MKCs lacking β1 integrin, which have high Rho activity, strong focal adhesions (nucleated through β6 integrin), and massive stress fibers (*Bandyopadhyay et al., 2012*; *Raghavan et al., 2003*). Further studies will be required to define why only engagement of β1 integrins is sufficient to exert high tension on the LINC complex.

How might tension on the nuclear lamina regulate epidermal differentiation? As a skin-specific mouse model lacking A- and B-type lamin expression also demonstrated precocious epidermal differentiation (*Jung et al., 2014*) and we find that MKC differentiation results in a relaxation of lamin A/C tension (*Figure 2D*, E), the LINC complex may act through tension-dependent remodeling of the nuclear lamina. The chromosome region housing the EDC moves away from the nuclear lamina during epidermal differentiation (*Gdula et al., 2013*; *Mardaryev et al., 2014*; *Williams et al., 2002*) and is associated with the loss of H3K27me3 chromatin marks on differentiation-specific genes

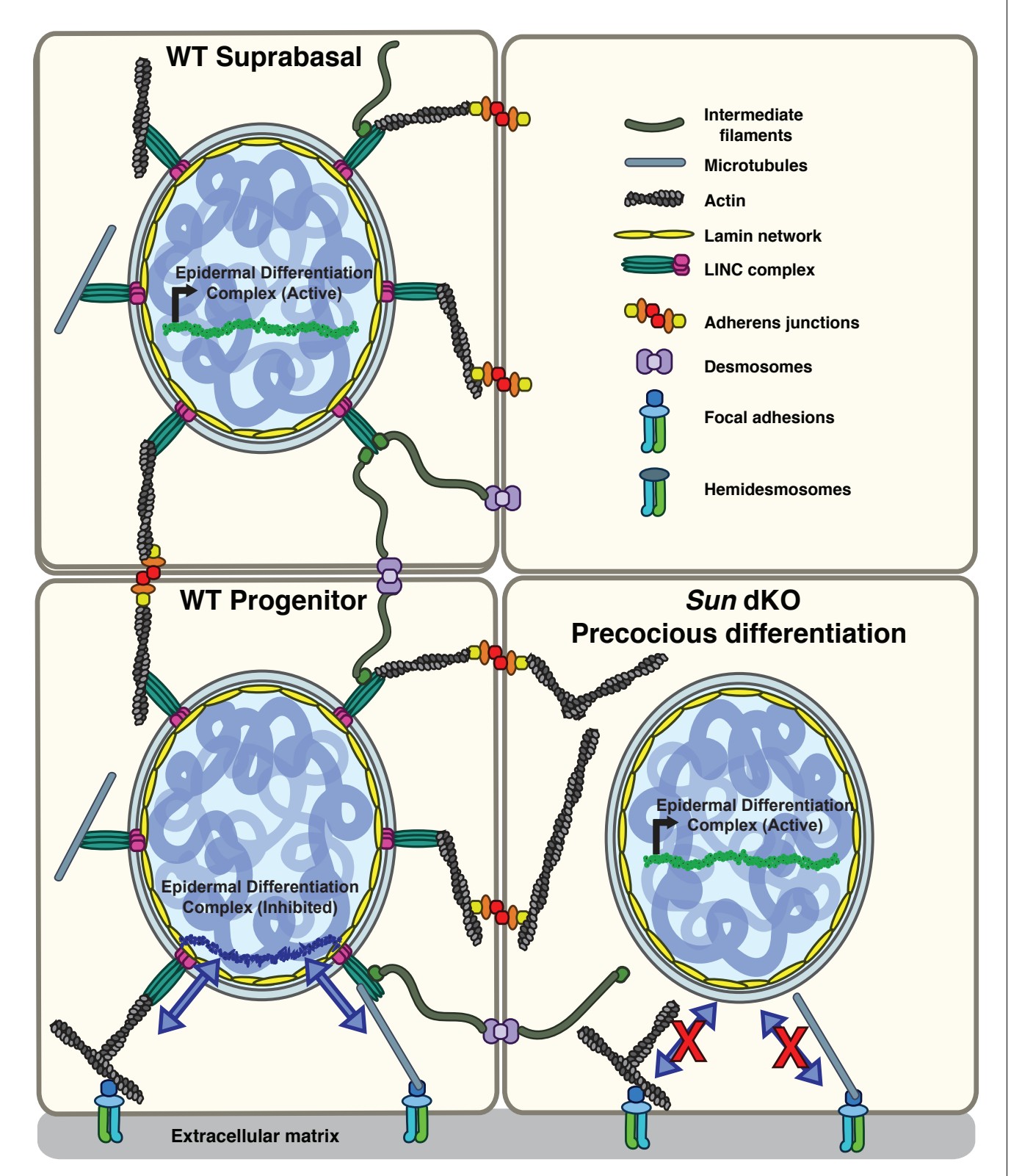

**Figure 6.** Model for the mechanism of precocious differentiation in the Sun dKO epidermis. WT progenitor cells adhere to the basal lamina through integrins, leading to tension on linker of nucleoskeleton and cytoskeleton (LINC) complexes that is transmitted to the basal surface of the nuclear lamina to maintain repression of the epidermal differentiation complex (EDC). As cells move into the suprabasal layer, they lose integrin engagement. In suprabasal cells that only engage cell-cell junctions, LINC complexes and the nuclear lamina are relaxed, leading to upregulation of EDC gene

*Figure 6 continued on next page*

*Figure 6 continued*

expression. In the Sun dKO epidermis, tension from engaged integrins is not propagated to the nucleus in progenitor cells despite normal adhesion, leading to precocious expression of EDC genes.

(*Lien et al., 2011*). If and how lamin tension influences this change in nuclear compartmentalization remains to be defined. We favor a model in which stretching of lamins drives remodeling of the composition of the nuclear lamina to maintain the epidermal progenitor state. Interestingly, the conformational epitope in lamin A/C examined in this work (*Figure 2*) overlaps with binding sites for DNA, histones, SUN proteins, and the integral inner nuclear membrane protein emerin (*Haque et al., 2010*; *Ihalainen et al., 2015*). Thus, the force-dependent differential exposure of these binding sites may influence the interaction of lamin A/C with the genome, either directly or through modulation of the activity of its binding partners. Indeed, stretching of MKCs leads to the release of emerin from the nuclear lamina; abrogating this mechanical response also leads to precocious differentiation (*Le et al., 2016*). Moreover, this response involves the formation of a perinuclear actin cage, a structure that we found previously to be perturbed in MKCs lacking SUN2 (*Stewart et al., 2015*). We note, however, that our ATAC-seq data do not reveal a global shift in chromatin accessibility in the absence and presence of LINC complex engagement, as observed previously in response to critical stretch of keratinocyte monolayers (*Le et al., 2016*; *Nava et al., 2020*), suggesting that constitutive tension on the LINC complex and the response to rapid stretching or cyclic stretch of the cell substratum may drive distinct responses.

Further work will be required to determine if the levels and geometry of forces exerted on LINC complexes differ between integrin-based adhesions and cell-cell adhesions. Another open question is whether building tension on the nuclear lamina requires a force opposing that is mediated by the LINC-actin-integrin axis. In our prior work, we found that control over nuclear position in MKCs at the periphery of cohesive colonies was dictated by a 'tug-of-war' between the actomyosin and microtubule networks (*Stewart et al., 2015*). Whether coupling of microtubules to LINC complexes contributes to building tension on the LINC-actin-integrin axis, and the magnitude of these intramolecular forces, remains an open question as all Nesprin TSMod constructs devised to date (*Arsenovic et al., 2016*; *Déjardin et al., 2020*), including the one we describe here, lack the spectrin repeats responsible for mediating interactions with microtubules and microtubule motors (*Schneider et al., 2011*). Examining whether the geometry of exogenous force application to cells influences nuclear lamina-dependent regulation of gene expression by modulating the extent of chromatin stretch (*Tajik et al., 2016*) represents another exciting direction.

# Materials and methods

## Key resources table

| Reagent type (species) or resource | Designation | Source or reference | Identifiers | Additional information |
|---|---|---|---|---|
| Strain, strain background (*Mus musculus*) | WT C57Bl/6J | Jackson Laboratories, Bar Harbor, ME | Stock number 000664 | |
| Genetic reagent (*M. musculus*) | *Sun1$^{-/-}$/Sun2$^{-/-}$*(Sun dKO) | Jackson Laboratories, Bar Harbor, ME | B6;129S6-*Sun1$^{tm1Mhan}$*/J Stock No: 012715 crossed to B6;129S6-*Sun2$^{tm1Mhan}$*/J Stock No: 012716 | |
| Cell line (*M. musculus*) | Primary WT keratinocyte | This paper | | Isolated from WT pups |
| Cell line (*M. musculus*) | Primary Sun dKO keratinocyte | This paper | | Isolated from Sun dKO pups |
| Cell line (*M. musculus*) | Integrin β1 null keratinocyte | *Bandyopadhyay et al., 2012* | | |

*Continued on next page*

*Continued*

| Reagent type (species) or resource | Designation | Source or reference | Identifiers | Additional information |
|---|---|---|---|---|
| Antibody | SUN1 antibody (rabbit monoclonal) | Abcam | ab124770 | IHC (1:1000) WB (1:100) |
| Antibody | SUN2 antibody (rabbit monoclonal) | Abcam | ab124916 | IHC (1:1000) WB (1:100) |
| Antibody | Keratin 10 antibody (rabbit polyclonal) | Gift from Julia Segre (*Harmon et al., 2013*) | | IHC (1:500) |
| Antibody | Keratin 1 antibody (chicken polyclonal) | Gift from Julia Segre *Harmon et al., 2013* | | IHC (1:500) |
| Antibody | Involucrin antibody (rabbit polyclonal) | Gift from Julia Segre *Harmon et al., 2013* | | IHC (1:500) |
| Antibody | Filaggrin antibody (chicken polyclonal) | Gift from Julia Segre *Harmon et al., 2013* | | IHC (1:500) |
| Antibody | β-Actin antibody (mouse monoclonal) | Abcam | ab13772 | WB: 1:1000 |
| Antibody | Conformationally sensitive Lamin A/C antibody | Abcam | ab8984 | IF: 1:200 |
| Recombinant DNA reagent | N2G-JM-TSMod (plasmid) | This paper | | Constructed from pEGFP-C1 containing mini-Nesprin-2G (*Luxton et al., 2010*) |
| Recombinant DNA reagent | N2G-JM-TSMod Dark Venus (plasmid) | This paper | | Mutation in Venus Y67L |
| Recombinant DNA reagent | N2G-JM-TSMod Dark mTFP (plasmid) | This paper | | Mutation in mTFP Y72L |
| Recombinant DNA reagent | NoT_TSMod (Plasmid) | This paper | | Constructed from pEGFP-C1 containing N2G-JM-TSMod |
| Sequence-based reagent | *Involucrin* RNA FISH probe | Thermo Fisher | VB1-3030396-VC | |
| Sequence-based reagent | *Sprr1b* RNA FISH probe | Thermo Fisher | VB4-3117172-VC | |
| Sequence-based reagent | GAPDH_F | This paper | qPCR primer | AGGTCGGTGT GAACGGATTTG |
| Sequence-based reagent | GAPDH_R | This paper | qPCR primer | TGTAGACCATGT AGTTGAGGTCA |
| Sequence-based reagent | Sprr1b_F | This paper | qPCR primer | GATCCCAGCGACCACAC |
| Sequence-based reagent | Sprr1b_R | This paper | qPCR primer | GCTGATGTGAA CTCATGCTTC |
| Sequence-based reagent | Sprr2d_F | This paper | qPCR primer | GTGGGCACAC AGGTGGAG |
| Sequence-based reagent | Sprr2d_R | This paper | qPCR primer | GCCGAGACTAC TTTGGAGAAC |
| Sequence-based reagent | Involucrin_F | This paper | qPCR primer | GCAGGAGAAG TAGATAGAG |
| Sequence-based reagent | Involucrin_R | This paper | qPCR primer | TTAAGGAAGT GTGGATGG |
| Sequence-based reagent | S100a14_F | This paper | qPCR primer | GGCAGGCTATAGGACA |

*Continued on next page*

Continued

| Reagent type (species) or resource | Designation | Source or reference | Identifiers | Additional information |
|---|---|---|---|---|
| Sequence-based reagent | S100a14_R | This paper | qPCR primer | CCTCAGCTCCGAGTAA |
| Peptide, recombinant protein | Fibronectin | Sigma-Aldrich | F4759 | 50 µg/mL |
| Peptide, recombinant protein | Poly-L-lysine | Sigma-Aldrich | P9155 | 50 µg/mL |
| Peptide, recombinant protein | Laminin | Thermo Fisher | CB-40232 | 50 µg/mL |
| Chemical compound, drug | Latrunculin A | Cayman Chemical Company | 10010630 | 0.5 µM |
| Commercial assay, kit | ViewRNA ISH Cell Assay kit | Thermo Fisher | QVC0001 | |
| Commercial assay, kit | Click-iTEdU Cell Proliferation Kit for Imaging | Invitrogen | C10337 | Alexa Fluor 488 dye |
| Commercial assay, kit | RNeasy Plus mini kit | QIAGEN | 74134 | |
| Commercial assay, kit | TruSeq RNA sample preparation kit | Illumina | RS-122-2001 | |
| Commercial assay, kit | iScript cDNA synthesis kit | Bio-Rad | 1708890 | |
| Commercial assay, kit | SYBR Green Supermix | Bio-Rad | 170-8882 | |
| Commercial assay, kit | Nextera Library Prep Kit | Illumina | 15028212 | |
| Commercial assay, kit | MinElute PCR Purification Kit | QIAGEN | 28004 | |
| Software, algorithm | ImageJ /Fiji | National Institutes of Health | | Version 1.50e |
| Software, algorithm | GraphPad Prism 8.0 | GraphPad | | Version 8.0 |
| Software, algorithm | PixFRET ImageJ Plugin | *Feige et al., 2005* | | |
| Software, algorithm | Gaussian fit | This paper | | https://github.com/LusKingLab/GaussianFit; *Carley, 2021*; copy archived at swh:1:rev:09e7545145b4dbbcb67d284a004d780176620130 |
| Software, algorithm | BowTie/TopHat2 | *Kim et al., 2013* | | |
| Software, algorithm | DESeq2 | *Love et al., 2014* | | |
| Software, algorithm | ENCODE ATAC-seq pipeline | *The ENCODE Project Consortium, 2013* Kundaje Lab | Version 1.8.0 | https://github.com/ENCODE-DCC/atac-seq-pipeline |
| Other | Prolong Gold with DAPI | Invitrogen | P36935 | |

*Continued on next page*

*Continued*

| Reagent type (species) or resource | Designation | Source or reference | Identifiers | Additional information |
|---|---|---|---|---|
| Other | Sera-Mag Select Beads | GE | 29343052 | |
| Other | CY 52–276 | Dow Corning | 52-276 | To make 3 kPa hydrogels |
| Other | Gil 2 Haematoxylin | Richard Allan Scientific | Cat # 72504 | |
| Other | Eosin-Y Alcoholic | Richard Allan Scientific | Cat # 71204 | |
| Other | JetPrime | Polyplus transfection | 114-07 | Transfection reagent |

## Cell culture and plasmid transfection

MKCs were isolated from skin from E18.5 embryos or newborn whole-body $Sun1^{-/-}/Sun2^{-/-}$ (Sun dKO) or WT pups as previously described (*Mertz et al., 2013*). Under sterile conditions, pups were sacrificed, and back skin was excised, washed in phosphate-buffered saline (PBS), and floated on dispase at 4°C for 16–20 hr. The epidermis was separated from the dermis with forceps and incubated in 0.25% trypsin for 15 min at room temperature (RT). Cells were liberated by trituration, filtered using a 40–70 μm strainer, and plated on mitomycin-C–treated J2 fibroblasts in medium-calcium medium (0.3 mM $CaCl_2$). After 2–4 passages, keratinocytes were plated on plastic dishes without feeder cells and maintained in media containing 0.05 mM $CaCl_2$ (E-low calcium media). Cells at low passage were stored under liquid nitrogen, and thawed cells were only used to passage 16. Integrin β1 null keratinocytes and corresponding WT controls (*Bandyopadhyay et al., 2012*) (kind gift from David Calderwood and Srikala Raghavan) were cultured at 32°C with 7.5% $CO_2$ in E media (DMEM/F12 in a 3:1 ratio with 15% fetal bovine serum supplemented with insulin, transferrin, hydrocortisone, cholera toxin, triidothyronone, and penicillin-streptomycin). MKCs were tested and found to be mycoplasma-free. Sun dKO MKC cell lines were validated by western blotting (see below).

## Construction and application of N2G-JM-TSMod constructs

The N2G-JM-TSMod construct was derived from pEGFP-C1 harboring mini-Nesprin2 (*Luxton et al., 2010*) (kind gift from Gant Luxton and Gregg Gundersen). The sequence encoding green fluorescent protein was removed by QuikChange mutagenesis. The mTFP-Venus tension sensor module (*Grashoff et al., 2010*) (kind gift from Martin Schwartz) was inserted just prior to the transmembrane domain. Dark controls were generated using QuikChange mutagenesis (Dark Venus: Y67L; Dark mTFP: Y72L). The no-tension (NoT_TSmod) construct was generated by inserting the mTFP-Venus tension sensor module 5′ of mini-Nesprin2 using the AgeI and XhoI restriction sites. For FRET experiments, cells were plated in E-low calcium media (or E media for Integrin β1 null MKCs ) 16 hr before transfections on glass-bottomed dishes (MatTek Corporation) coated with 50 μg/mL fibronectin (20 min at room temperature, Sigma-Aldrich), poly-L-lysine (2 hr at room temperature, Sigma P9155), or laminin (2 hr at 37°C, Thermo, Cat# CB-40232). Cells were transfected using JetPrime reagent (Polyplus) according to the manufacturer's instructions and imaged the following day. Donor bleedthrough samples consisted of cells transfected with mini-Nesprin2G-mTFP1 alone (*Figure 1—figure supplement 1*). Acceptor cross-excitation samples consisted of cells transfected with mini-Nesprin2G-Venus alone (*Figure 1—figure supplement 1*). In addition, untransfected cells were also used as dark controls (*Figure 1—figure supplement 1*). FRET samples consisted of Nesprin2G-JM-TSMod. Actin depolymerization was achieved by the addition of 0.5 μM Lat A (or vehicle control) for 5 hr prior to imaging.

## Generation of hydrogels

The soft 3 kPa PDMS gels were prepared by mixing a 1:1 ratio (w/w) of CY 52-276A and CY 52-276B (Dow Corning). Components were mixed thoroughly and immediately degassed for 15 min in a desiccator. The surface of 35 mm glass-bottomed culture dishes (MatTek Corporation) were covered

with 200 µL of the mixed gel. Each dish was then spin-coated at 1000 rpm for 60 s using a spin coater instrument (Headway Research, PWM32). Gels were cured at room temperature overnight. The next morning gels were sterilized for 30 min under UV light and then washed once with sterile PBS. Rheometry analysis (ARES-LS1) confirmed that the Young's modulus of the PDMS was ~3 kPa.

## RNA FISH experiments

WT and Sun dKO MKCs were plated at low density (15,000 cells/well of 24-well dish) onto glass coverslips coated with 50 µg/mL fibronectin (Sigma-Aldrich), MKCs were cultured in E-low calcium media overnight before switching to 1.2 mM calcium media to induce differentiation for 24 or 48 hr. RNA FISH was performed using the ViewRNA ISH Cell Assay kit (Thermo Fisher Cat# QVC0001) with probes for *involucrin* (Thermo Fisher Cat# VB1-3030396-VC) and *Sprr1b* (Thermo Fisher Cat# VB4-3117172-VC) according to manufacturer's instructions. Coverslips were mounted onto glass slides with Prolong Gold with DAPI (Invitrogen P36935) and sealed with clear nail polish. Images were acquired with the Zeiss Imager M1 using Zen software. Images for the same marker were acquired at the same exposure, pixel range, and gamma values. Acquired images were equally brightened, contrasted, and cropped using ImageJ/Fiji (version 1.50e) software (*Schindelin et al., 2012*). For quantitation, signal threshold was determined using no probe controls for each individual experiment. Colony outlines were drawn and then cells were counted and scored as positive or negative for gene expression and interior or periphery based on location within colony. Statistical analysis was performed using Prism 8 software.

## Mouse tissue isolation, histology, and immunofluorescence staining

This study was performed in strict accordance with the recommendations in the Guide for the Care and Use of Laboratory Animals of the National Institutes of Health (IACUC protocol number 2018-11248). All animal care and experimental procedures were conducted in accord with requirements approved by the Institutional Animal Care and Use Committee of Yale University. At harvest, embryos were submerged in O.C.T. compound (Tissue-Tek) and sectioned using a cryostat (CM3050S; Leica). Sections were cut in a specific and consistent orientation relative to embryo morphology and stained with hematoxylin and eosin for routine histopathology or incubated with primary antibodies and Alexa Fluor-conjugated secondary antibodies for indirect immunofluorescence as previously described (*Stewart et al., 2015*). Primary antibodies against SUN1 (1:100, Abcam, ab124770) and SUN2 (1:100; Abcam ab124916) were used. Primary antibodies to keratin 10 (K10), keratin 1 (K1), involucrin, and filaggrin were gifts from the Segre lab and are described in *Harmon et al., 2013*. Images were acquired with the Zeiss Imager M1 using Zen software. Images for the same marker were acquired at the same exposure, pixel range, and gamma values. Acquired images were equally brightened, contrasted, and cropped using ImageJ/Fiji. Spinous layer thickness was quantitated by making 5–8 measurements per image using ImageJ software. Statistical analysis was performed using Prism 8 software.

## Western blot analysis

Whole-cell lysates of WT, *Sun2-/-*, and Sun dKO MKCs were prepped as previously described (*Stewart et al., 2015*). Primary antibodies against SUN1 (1:100, Abcam, ab124770), SUN2 (1:100; Abcam ab124916), and β-actin (1:1000; mouse; Abcam) were used.

## EdU incorporation and quantitation

Pregnant females were pulsed with EdU via IP injection when embryos were age E14.5. 24 hr after injection, embryos were isolated and embedded in O.C.T. media (Tissue-Tek) and stored at −80°C until sectioning using a cryostat (CM3050S; Leica). After sectioning, the Click-iTEdU Cell Proliferation Kit for Imaging, Alexa Fluor 488 dye (Invitrogen C10337), was used according to manufacturer's instructions to evaluate EdU incorporation. Tissue sections were then co-stained for K10 (see 'Mouse tissue isolation, histology, and immunofluorescence staining'). Before imaging, coverglass was mounted onto slides using Prolong Gold with DAPI (Invitrogen P36935) and sealed with clear nail polish. Images were acquired with the Zeiss Imager M1 using Zen software. Total EdU-positive cells were counted and normalized to the total number of epidermal cells (determined by DAPI staining). Location of proliferating cells was determined by K10 staining (basal keratinocytes [EdU+/K10-],

suprabasal keratinocytes [EdU+/K10+]) and then normalized to total EdU-positive cells. Statistical analysis was performed using Prism 8 software.

## Electron microscopy

Transmission electron microscopy was performed in the Yale School of Medicine Center for Cellular and Molecular Imaging Electron Microscopy core facility. Back skin sections from *Sun 1+/-/Sun2-/-* and Sun dKO pups were isolated at age P0.5; three mice were examined for each genotype. Tissue blocks were fixed in 2.5% glutaraldehyde/2% paraformaldehyde in 0.1 M sodium cacodylate buffer, pH 7.4, for 30 min at RT and 1.5 hr at 4°C. The samples were rinsed in sodium cacodylate buffer and were postfixed in 1% osmium tetroxide for 1 hr. The samples were rinsed and en bloc stained in aqueous 2% uranyl acetate for 1 hr followed by rinsing, dehydrating in an ethanol series to 100%, rinsing in 100% propylene oxide, infiltrating with EMbed 812 (Electron Microscopy Sciences) resin, and baking overnight at 60°C. Hardened blocks were cut using an ultramicrotome (UltraCut UC7; Leica). Ultrathin 60 nm sections were collected and stained using 2% uranyl acetate and lead citrate for transmission microscopy. Carbon-coated grids were viewed on a transmission electron microscope (TecnaiBioTWIN; FEI) at 80 kV. Images were taken using a CCD camera (Morada; Olympus) and iTEM (Olympus) software. The length and number of hemidesmosomes was quantitated from sections from three mice of each genotype.

## Immunofluorescence

WT and Sun dKO keratinocytes were plated at low density (15,000 cells/well of 24-well dish) onto glass coverslips coated with 50 μg/mL fibronectin (Sigma-Aldrich), MKCs were cultured in E-low calcium media overnight before switching to 1.2 mM calcium media to induce differentiation for 48 hr. Cells were fixed with methanol at −20°C for 5 min and washed with PBS. Cells were permeabilized using 0.5% Triton X-100 in PBS at RT for 20 min and blocked using 10% goat serum, 5% bovine serum albumin (BSA), and 0.5% Tween 20 in PBS for 1 hr. Cells were incubated in primary antibody, conformationally sensitive lamin A/C (1:200; ab8984; Abcam), diluted in blocking buffer at 4°C overnight. Coverslips were washed with PBS for three 5 min intervals and incubated with Alexa Fluor 488-conjugated secondary antibody (1:1000; Invitrogen Cat# A-11029) diluted in blocking buffer at RT for 1 hr. Coverslips were then costained with Hoechst 33342 (1:2000; Thermo Fisher Scientific) and Alexa Fluor 594-conjugated Wheat Germ Agglutinin (1:1000; Life Technologies) diluted in PBS for 5 min at RT. Coverslips were washed with PBS, mounted using Fluoromount-G, and sealed with clear nail polish.

## Imaging and image analysis

Live FRET imaging was performed on a Zeiss LSM 710 DUO NLO confocal microscope using a 100×, 1.4 NA oil objective with a stage maintained at 37°C and 7.5% $CO_2$. Images were acquired using Zen software. FRET imaging was performed in a similar manner as previously described (*Grashoff et al., 2010*; *Kumar et al., 2016*). Three sequential images were acquired: the donor mTFP1 channel using a 458 nm laser line (ex), 458 nm MBS filter, and PMT detector set for mTFP1 emission; the acceptor Venus channel using a 514 nm laser line (ex), 458/514 nm MBS filter, and PMT detector set for Venus emission; and the FRET channel using a 458 nm laser line (ex), 458/514 nm MBS filter, and PMT detector set for Venus emission. In all cases, the nuclear midplane was imaged. Acquisition settings were standardized and maintained during experiments. FRET image analysis was performed using the intensity-based FRET method, implemented as previously described (*Kumar et al., 2016*). Nonlinear spectral bleed-through corrections were first determined using the PixFRET plugin (*Feige et al., 2005*) for ImageJ/Fiji (*Schindelin et al., 2012*). Donor mTFP1 leakage was quantified using cells transfected with mini-Nesprin2G-mTFP1 alone (dark Venus control), while acceptor Venus cross-excitation was quantified using cells transfected with mini-Nesprin2G-Venus alone (dark mTFP control). At least 10 cells each were used for bleed-through corrections. For FRET index determination, masks consisting of a three pixel-wide band encompassing the nuclear envelope were used to segment the nuclear envelope. Mean FRET index per nucleus was then determined using previously published software, either PixFRET (*Figure 1*; *Feige et al., 2005*) or in-house MATLAB scripts (*Figure 2*; *Kumar et al., 2016*).

Imaging of conformationally sensitive lamin A/C was performed on a Leica SP5 confocal microscope using the LAS-AF software. Lamin A/C (AF488) was imaged using a 488 nm laser line (ex) and PMT detector set for AF488 emission; and the WGA (AF594) using a 594 nm laser line (ex) and PMT detector set for AF594 emission. PMT gain was adjusted for each sample to avoid over-/under-exposure. Acquisition settings were standardized and maintained during experiments. Single-plane images to determine the location of each cell within a colony were acquired using a 40× 1.25 NA air objective, imaging the AF488 and AF594 channels sequentially. Next, Z-stacks of each images were acquired using a 63×, 1.4 NA oil objective in the AF488 channel. Image zoom and size was adjusted such that voxel size was 55–65 nm in the xy-plane and 130 nm in the z-plane. Image stacks were deconvolved using Huygens Professional software (Scientific Volume Imaging, The Netherlands, http://svi.nl) as described (*Ihalainen et al., 2015*). A theoretical point spread function was used for iterative deconvolution. Deconvolution was performed using the following software parameters: image signal to noise was set to 5, the quality threshold was 0.01, and maximum iterations was 50 (however, usually fewer than 20 iterations were required to reach the quality threshold). Images were then analyzed using ImageJ/Fiji (*Schindelin et al., 2012*). The apical-basal lamin A/C intensity was measured for a single XZ or YZ slice from the middle of each individual nucleus of interest. The intensity of antibody staining was measured from the apical to the basal side of the nucleus using a straight line along the z-axis that is approximately half the width of the nucleus of interest. The fluorescence intensity across the z-axis was plotted and fit to two gaussian curves, one corresponding to the apical side of the nuclear envelope and the second corresponding to the basal side. The area under each curve and the ratio of these areas were calculated using MATLAB R2019b, and the software can be found on GitHub at: https://github.com/LusKingLab/GaussianFit.

## RNAseq

WT and Sun dKO cells were grown in E-low calcium media for 24 hr to 100% confluency (undifferentiated) or were switched to high calcium media to induce adhesion formation and differentiation for 48 hr. Total RNA was isolated using the RNeasy Plus kit (QIAGEN) according to the manufacturer's instructions for three biological replicates for each condition. cDNA was synthesized using reagents from the TruSeq RNA sample preparation kit (Illumina) according to the manufacturer's instructions. cDNA libraries were sequenced (paired end 75 nts) at the Yale Stem Cell Center Genomics and Bioinformatics Core on the HiSeq4000 platform. Reads were mapped using BowTie/TopHat2 (*Kim et al., 2013*) to the mm10 genome build. Differentially expressed genes between the WT and Sun dKO conditions in the undifferentiated and differentiated states were identified using DESeq2 (*Love et al., 2014*). All sequencing data can be accessed at NCBI under BioProject Accession PRJNA636991.

## RT-qPCR

WT and Sun dKO MKCs were plated in a fibronectin-coated (50 ng/mL; Sigma-Aldrich) 6-well dish such that they were 70–80% confluent. They were cultured in E-low calcium media overnight before switching to 1.2 mM calcium media to induce differentiation for 48 hr. Total RNA was isolated using the RNeasy Plus kit (QIAGEN) according to the manufacturer's instructions. The iScript cDNA Synthesis Kit (Bio-Rad) was used to generate cDNA from equal amounts of total RNA (1 mg) according to the manufacturer's instructions. Quantitative real-time PCR was performed with a Bio-Rad CFX96 using iTaq Universal SYBR Green Supermix (Bio-Rad) for 40 cycles. Primers used include GAPDH forward: AGGTCGGTGTGAACGGATTTG and reverse: TGTAGACCATGTAGTTGAGGTCA; Sprr1b forward: GATCCCAGCGACCACAC and reverse: GCTGATGTGAACTCATGCTTC; Sprr2d forward: GTGGGCACACAGGTGGAG and reverse: GCCGAGACTACTTTGGAGAAC; involucrin forward: GCAGGAGAAGTAGATAGAG and reverse: TTAAGGAAGTGTGGATGG; S100a14 forward: GGCAGGCTATAGGACA and reverse: CCTCAGCTCCGAGTAA. PCR product levels were normalized to GAPDH mRNA levels.

## ATAC-seq

ATAC-seq libraries were generated as previously described (*Buenrostro et al., 2015*). Briefly, 50,000 nuclei were purified from WT or Sun dKO cells and DNA was tagmented using the Nextera Library Prep Kit (Illumina). Tagmented DNA was purified using the QIAGEN MinElute PCR

Purification kit, then amplified using PCR. RT-qPCR was used to determine the appropriate number of cycles of amplification. The DNA was then purified twice using Sera-Mag Select beads (GE). Library purity and quality was assessed using an Agilent Bioanalyzer, then was sequenced on the Illumina HiSeq 4000 using paired end reads. Sequencing was performed at the Yale Stem Cell Center Genomics Core facility. Sequences were processed using the ENCODE ATAC-seq pipeline version 1.8.0, an automated end-to-end quality control and processor of ATAC-seq data. The minimum template seeded the input JSON onto the mm10 genome build with the autodetect adapter parameter set to true. All sequencing data can be accessed at NCBI under BioProject Accession PRJNA636991.

## Acknowledgements

We thank the laboratories of GW Gant Luxton (University of California, Davis), Gregg Gundersen (Columbia University), Julia Segre (NHGRI), Srikala Raghavan (Institute for Stem Cell Science and Regenerative Medicine), David Calderwood (Yale University), Martin Schwartz, (Yale University), and Chinedum Osuji (Yale University) for sharing of reagents, instrumentation, and/or expertise. The high-throughput sequencing was conducted by the Yale Stem Cell Center Genomics and Bioinformatics Core facility, which was supported by the Connecticut Regenerative Medicine Research Fund. This work would not have been possible without the support of the Ludwig Family Foundation and the Physical Engineering Biology Program at Yale University.

## Additional information

### Competing interests

Megan C King: Reviewing editor, *eLife*. Valerie Horsley: Reviewing editor, *eLife*. The other authors declare that no competing interests exist.

### Funding

| Funder | Grant reference number | Author |
| --- | --- | --- |
| National Institutes of Health | R01 GM129308 | Emma Carley<br>Iman Jalilian<br>Megan C King |
| American Heart Association | 16PRE27460000 | Rachel M Stewart |
| Ludwig Family Foundation | | Megan C King<br>Rachel M Stewart |
| National Institutes of Health | R01 AR060295 | Valerie Horsley |
| National Institutes of Health | R01 AR069550 | Valerie Horsley |
| National Institutes of Health | T32 AR007016 | Abigail Zieman |
| National Institutes of Health | T32 GM007223 | Emma Carley |

The funders had no role in study design, data collection and interpretation, or the decision to submit the work for publication.

### Author contributions

Emma Carley, Data curation, Software, Investigation, Visualization, Methodology, Writing - review and editing; Rachel M Stewart, Conceptualization, Data curation, Formal analysis, Funding acquisition, Validation, Investigation, Visualization, Methodology, Writing - original draft; Abigail Zieman, Data curation, Formal analysis, Funding acquisition, Validation, Investigation, Visualization, Methodology, Writing - review and editing; Iman Jalilian, Formal analysis, Investigation, Visualization, Methodology, Writing - review and editing; Diane E King, Data curation, Software, Formal analysis; Amanda Zubek, Conceptualization, Formal analysis, Investigation, Visualization, Methodology; Samantha Lin, Investigation; Valerie Horsley, Conceptualization, Resources, Data curation, Formal analysis, Supervision, Funding acquisition, Validation, Writing - original draft, Project administration; Megan C King, Conceptualization, Resources, Data curation, Formal analysis, Supervision, Funding

acquisition, Validation, Visualization, Writing - original draft, Project administration, Writing - review and editing

### Author ORCIDs
Abigail Zieman (iD) https://orcid.org/0000-0001-8236-207X
Valerie Horsley (iD) https://orcid.org/0000-0002-1254-5839
Megan C King (iD) https://orcid.org/0000-0002-1688-2226

### Ethics

Animal experimentation: This study was performed in strict accordance with the recommendations in the Guide for the Care and Use of Laboratory Animals of the National Institutes of Health. All animal care and experimental procedures were conducted in accord with requirements approved by the Institutional Animal Care and Use Committee of Yale University. IACUC Approval 2018-11248.

### Decision letter and Author response

Decision letter https://doi.org/10.7554/eLife.58541.sa1
Author response https://doi.org/10.7554/eLife.58541.sa2

## Additional files

### Supplementary files

• Supplementary file 1. Table of RNAseq (Tab 1) and GO Term analysis (Tab 2) for Sun dKO versus WT mouse keratinocytes grown in low calcium media (undifferentiated).

• Supplementary file 2. Table of RNAseq (Tab 1) and GO Term analysis (Tab 2) for Sun dKO versus WT mouse keratinocytes grown in high calcium media (differentiated).

• Supplementary file 3. Quality control metrics for assay for transposase-accessible chromatin using sequencing (ATAC-seq) experiments.

• Supplementary file 4. Annotated genes that demonstrate differential chromatin accessibility between WT and Sun dKO mouse keratinocytes (MKCs) as assessed by assay for transposase-accessible chromatin using sequencing (ATAC-seq). Tab 1 lists genes for which ATAC-seq peaks are present in WT but absent in Sun dKO MKCs, and Tab 2 lists genes for which ATAC-seq peaks are present in Sun dKO but absent in WT MKCs.

• Supplementary file 5. Focused analysis of assay for transposase-accessible chromatin using sequencing (ATAC-seq) changes between WT and Sun dKO mouse keratinocytes for additional epidermal differentiation genes and genes tied to proliferation of keratinocyte progenitors.

• Transparent reporting form

### Data availability

Sequencing data have been deposited as a single BioProject at NCBI with accession number PRJNA636991.

The following dataset was generated:

| Author(s) | Year | Dataset title | Dataset URL | Database and Identifier |
|---|---|---|---|---|
| Carley E, Stewart RK, Zieman A, Jalilian I, King DE, Zubek A, Lin S, Horsley V, King MC | 2020 | Regulation of epidermal differentiation by the LINC complex | https://www.ncbi.nlm.nih.gov/bioproject/PRJNA636991/ | NCBI BioProject, PRJNA636991 |

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
