## [Decision Letter]

**Acceptance summary:**

This work will be of broad interest, because it provides new insights into how cells sense tension and transmit it to the cell nucleus. With ever increasing excitement as to how cells sense tension and transmit it to the nucleus, your new work will now be an important addition to this arena.

**Decision letter after peer review:**

Thank you for submitting your article "The LINC complex transmits integrin-dependent tension to the nuclear lamina and represses epidermal differentiation" for consideration by *eLife*. Your article has been reviewed by 3 peer reviewers, one of whom is a member of our Board of Reviewing Editors, and the evaluation has been overseen by Anna Akhmanova as the Senior Editor. The reviewers have opted to remain anonymous.

The reviewers have discussed the reviews with one another and the Reviewing Editor has drafted this decision to help you prepare a revised submission.

Summary:

The reviews of your manuscript are now in. As you can see, the consensus is that your manuscript strives to address a fundamental and largely understudied question, and your tools and strategy have the potential to significantly advance our understanding of how connections between cell-matrix adhesions and the nucleus have direct consequences on cell fate. That said, all your reviewers encourage you to provide better support for the central claims of your manuscript. There are five pressing concerns summarized below.

Essential revisions:

1. Their central hypothesis is that a relay of tension between integrins and the LINC complex maintains nuclear organization such that EDC gene expression is repressed. As tension on basally localized integrins is released, either by delamination or by redistribution of forces to cell-cell junctions (as happens in the middle of keratinocyte colonies), the chromatin state changes such that transcription of the differentiation program may be de-repressed. This attractive hypothesis needs to be tested more rigorously using their current models and existing experimental tools. For example, DNA-FISH approaches (as used in Mardaryev, Development, 2014) to test whether the positioning of the EDC chromosomal region within the nucleus is differentially located the interior of colonies (where tension is low) compared to the outside of colonies (where tension is high), or spatially correlates with LINC tension at the single-cell level (if their tension sensor is suitable for fixation and DNA-FISH conditions). The authors should also test whether nuclear positioning of the EDC cluster and/or genes associated with epidermal differentiation is misregulated in the Sun1/2-dKO epidermis. The inclusion of such data would significantly strengthen the central claim and broader impact of the manuscript.

2. There is a paucity of validation for the FRET sensor construct. To make conclusions about tension, a number of experiments can be conducted. A version of the construct lacking the actin-binding or KASH domains of Nesprin2 is important; similar controls were vital to the conclusions drawn in the original Schwartz paper (Grashoff et al., 2010) and in other contexts where the TS-module has been used (eg. Cai et al., Cell, 2013). Having the TS "free floating" on the N-terminus would also be valuable. These controls are also very important as e.g. treatment with latruculin or other inhibitors can affect the FRET efficiency in similar directions independent of tension. Other manipulations of cell tension (besides poly-lysine coating and Latrunculin treatment), such as coating with varying concentrations of fibronectin, culturing cells on substrates of different stiffness, or exogenous force application to culture substrates, would significantly improve confidence that their construct indeed reports on mechanical tension in the nuclear envelope. Standard for validation of FRET experiments is to include an acceptor photobleaching experiment. Also, although the raw data for Venus/mTFP channels and bleedthrough controls with single colour constructs should be shown.

3. The data with the force-sensitive epitope LaminA/C antibody is not of sufficient quality. In the original paper where this tool was developed (Nat Materials, 2015), the signal of the LaminA/C was normalized against signal from a force-insensitive LaminB antibody. Otherwise, how do we know we are looking at the nuclear envelope? Or that there aren't overall differences in nuclear envelope morphology/composition that contribute to the differences in A/C signal that we see, rather than an effect that can be specifically attributed to tensional differences?

4. Some baseline description of the LINC complex in the epidermis would provide insights that may lend support for (or refute) their hypothesis. For example, where in the epidermis is Sun1/2 expressed? Is there any developmental regulation of LINC complex organization? One might predict reduced expression or altered localization of LINC complex components (such as SUN1/2) in the differentiated layers of the epidermis. Can the force-sensitive Lamin-A/C antibody be used in vivo to see if there are differences between staining patterns in the basal layer where cells have integrin-mediated contacts compared to the suprabasal, differentiating layers where they do not? Are there differences in nuclear morphology in the presence or absence of sun1/2 dKO epidermis that might support their model? Does loss of Sun1/Sun2 alter nuclear stiffness? If the authors can show that the regulation of molecular tension across the Linc complex directly correlates to changes in nuclear tension, this will strengthen the paper.

5. The molecular links how integrins control tension on the Linc complex, or how altered tension at the Linc complex controls differentiation are not at all addressed, and the paper should at least provide some insight into either upstream or downstream control of how integrin dependent regulation of Sun1/2 regulates differentiation; e.g. explore the molecular link between integrin and the linc complex (in their model both actin and MTs are potentially implicated for example), alterations in epigenetic marks, chromatin organization of e.g. relocalization of the EDC locus etc.

The conditional b1 and KO of integrin a3 integrins do not result in altered/precocious differentiation in the skin (although it does affect proliferation and hair follicle morphogenesis). Presumably the link between integrins and the LINC complex under these conditions would be altered as well? The idea that LINC complex may control differentiation is possible, however the role of integrin based adhesions in this process has not been parsed out. At the very least the authors could have disrupted integrin signaling using KO cells or blocking antibodies. So it is possible that there may be alternative mechanisms? This should be considered.

[Editors' note: further revisions were suggested prior to acceptance, as described below.]

Thank you for submitting your revised article "The LINC complex transmits integrin-dependent tension to the nuclear lamina and represses epidermal differentiation" for consideration by *eLife*. Your article has now been re-reviewed by a Reviewing Editor and by Anna Akhmanova as the Senior Editor.

We agree that your manuscript is much improved and has an important message for *eLife*. That said, there still several items remaining, that we feel need to be addressed prior to publication.

Three issues remain. First, changes in nuclear structure and likely mechanics in keratinocytes induce rather global changes in chromatin and transcription (Le et al., 2016), and hence it would be helpful if you can present the ATAC data more generally and not just pick a few loci to analyze. Are your data specific for the EDC locus or are other differentiation genes also affected? One would also predict changes in the so-called stem cell genes-can you comment on whether this is observed? These minor adjustments will provide even a better global vision on how the nuclear envelope tension regulates gene expression and differentiation.

Second, the b1 KO cells have to be cultured in high calcium media and have very high levels of Rho activity, strong focal adhesions (nucleated through integrin b6) and massive stress fibers (Raghavan et al. 2003, Bandyopadhyay and Raghavan 2012). Given the role of the actin cytoskeletal network in transmitting the forces via the LINC complex to the nucleus, it seems surprising that the tension sensor shows such high FRET (low tension) compared to the WT cells. It would be helpful if you could add a bit of discussion as to how you interpret these data and reconcile this point. Toning down some of the conclusions made regarding this point might be warranted.

Finally, it would seem that you should add a brief discussion and referencing of the recent Nava et al. Cell paper, that shows heterochromatin driven changes in nuclear mechanics in keratinocytes.

---

## [Author Response]

Essential revisions:1. Their central hypothesis is that a relay of tension between integrins and the LINC complex maintains nuclear organization such that EDC gene expression is repressed. As tension on basally localized integrins is released, either by delamination or by redistribution of forces to cell-cell junctions (as happens in the middle of keratinocyte colonies), the chromatin state changes such that transcription of the differentiation program may be de-repressed. This attractive hypothesis needs to be tested more rigorously using their current models and existing experimental tools. For example, DNA-FISH approaches (as used in Mardaryev, Development, 2014) to test whether the positioning of the EDC chromosomal region within the nucleus is differentially located the interior of colonies (where tension is low) compared to the outside of colonies (where tension is high), or spatially correlates with LINC tension at the single-cell level (if their tension sensor is suitable for fixation and DNA-FISH conditions). The authors should also test whether nuclear positioning of the EDC cluster and/or genes associated with epidermal differentiation is misregulated in the Sun1/2-dKO epidermis. The inclusion of such data would significantly strengthen the central claim and broader impact of the manuscript.

We have carried out a variety of additional experiments to provide more direct evidence for (1) changes at the level of the chromatin at the EDC upon loss of the LINC complex and (2) the ability of disrupting β1-integrin to release tension on the LINC complex and recapitulate loss of the LINC complex with respect to regulation of epidermal differentiation genes.

Specifically, in this revision we added:

1) Additional analysis of individual EDC genes in WT and *Sun* dKO MKCs (Figure 4—figure supplement 1).

2) Direct assessment of MKCs lacking β1-integrin to demonstrate that β1-integrin is essential to 1) drive the high tension state on the LINC complex (Figure 1 J, K) and 2) repress epidermal differentiation markers, thereby recapitulating our observations in *Sun dKO* MKCs (Figure 4I).

3) We believe the most compelling addition are our results from Assay for Transposase Accessible Chromatin followed by sequencing (ATAC-seq) in WT and *Sun* dKO mouse keratinocytes (MKCs)(new Figure 5). We observe a clear increase in chromatin accessibility as revealed by gains in ATAC peaks in the EDC of *Sun dKO* MKCs across the EDC in the low calcium condition when these genes are repressed in WT MKCs, indicative of a more open chromatin state and consistent with our model of precocious differentiation.

We tried extensively to analyze the EDC position by FISH as suggested. Although we were able to detect the EDC at a central nuclear position in *Sun* dKO MKCs, we were not able to robustly detect signal in WT MKCs for comparison. We believe this could reflect our inability to sufficiently denature the probe targets when the region is heterochromatized, as we expect it to be in WT MKCs cultured in low calcium media. Despite our inability to complete this experiment, the new data described above provide additional support for the role of the LINC complex in regulating chromatin accessibility at epidermal differentiation genes and their expression.

2. There is a paucity of validation for the FRET sensor construct. To make conclusions about tension, a number of experiments can be conducted. A version of the construct lacking the actin-binding or KASH domains of Nesprin2 is important; similar controls were vital to the conclusions drawn in the original Schwartz paper (Grashoff et al., 2010) and in other contexts where the TS-module has been used (eg. Cai et al., Cell, 2013). Having the TS "free floating" on the N-terminus would also be valuable. These controls are also very important as e.g. treatment with latruculin or other inhibitors can affect the FRET efficiency in similar directions independent of tension. Other manipulations of cell tension (besides poly-lysine coating and Latrunculin treatment), such as coating with varying concentrations of fibronectin, culturing cells on substrates of different stiffness, or exogenous force application to culture substrates, would significantly improve confidence that their construct indeed reports on mechanical tension in the nuclear envelope. Standard for validation of FRET experiments is to include an acceptor photobleaching experiment. Also, although the raw data for Venus/mTFP channels and bleedthrough controls with single colour constructs should be shown.

We thank the reviewers for their feedback and suggestions. We have provided additional controls to ensure that changes in the FRET indexes of the mini-Nesprin2 TSMod (N2G-JM-TSMod) indeed reflect tension on the LINC complex. To this end, we made a tension insensitive construct (no-tension construct) by inserting the TSMod at the N-terminus of the mini-Nesprin2 (Figure 1 A, “NoT_TSMod”) where it cannot bear any load. Comparing FRET index measurements at the nuclear envelope for MKCs transfected with the TSMod inserted at the juxtamembrane region (N2G-JM-TSMod) and the NoT_TSMod control (Figure 1B,C) indeed revealed that the N2G-JM-TSMod is under tension in WT MKCs. We also compared the response of the two TSMods to treatment with latrunculin A to depolymerize actin. While the N2G-JM-TSMod was under higher tension that was relaxed upon treatment of latrunculin A (Figure D,E), the tension insensitive NoT_TSMod control showed no change in tension at the nuclear envelope (Figure 1—figure supplement 1B). Last, we provide new evidence that tension on the N2G-JM-TSMod is sensitive to the mechanics of the substrate (Figure 1F,G). Bolstered by these additional data we conclude that (1) the TSMod is sensitive to actin-dependent tension at the nuclear envelope; and (2) the changes in FRET efficiency reported by the N2G-JM-TSMod reflects intramolecular tension exerted on the juxtamembrane region, as the no tension control is insensitive to load. As suggested by the reviewers, we also now show additional experimental controls including “dark” donor only and acceptor only constructs used for bleed through correction (Figure 1—figure supplement 1A,C,D).

3. The data with the force-sensitive epitope LaminA/C antibody is not of sufficient quality. In the original paper where this tool was developed (Nat Materials, 2015), the signal of the LaminA/C was normalized against signal from a force-insensitive LaminB antibody. Otherwise, how do we know we are looking at the nuclear envelope? Or that there aren't overall differences in nuclear envelope morphology/composition that contribute to the differences in A/C signal that we see, rather than an effect that can be specifically attributed to tensional differences?

We thank the reviewers for this critical feedback. We agree that ideally we would co-stain nuclei with a force-insensitive lamin A/C or lamin B antibody to address the concerns raised by the reviewers. However, we have tested numerous commercially available tension insensitive lamin antibodies, including those used in the 2015 Nature Materials paper, as well as antibodies to other nuclear envelope and nuclear pore complex proteins. Ultimately, we were unable to identify an antibody that stained the nuclear envelope of the mouse keratinocytes under conditions compatible with the tension-sensitive lamin A/C antibody with the performance required for this experiment (although we verified this does work well in fibroblasts, as published). Since we found that we could successfully co-stain human epidermal keratinocytes, we further attempted to validate our system in this model. However, the nuclei of these cells are flatter than the mouse keratinocyte nuclei, and we were unable to achieve sufficient resolution with our confocal imaging and deconvolution set-up to resolve the apical and basal surfaces of the nucleus, even with manipulation of substrate conditions. This said, we developed our approach specifically because it allows us to robustly resolve the nuclear and basal surfaces. We have softened the language we use in the revised manuscript to acknowledge that while we observe decreased staining with this conformationally-sensitive antibody, this could reflect multiple changes in the state of the basal lamina, including a higher tension state.

The revised manuscript also includes abundant further characterization of the our N2G-JM-TSMod including its sensitivity to the presence of β1 integrin (Figure 1J,K) and the stiffness of the substrate on which the MKCs are plated (Figure 1F,G). As our observations using the tension-sensitive lamin A/C antibody mirror the results obtained from our now thoroughly-characterized N2G-JM-TSMod, we continue to favor the model that LINC complexes are under more tension in cells at the colony periphery versus the interior.

4. Some baseline description of the LINC complex in the epidermis would provide insights that may lend support for (or refute) their hypothesis. For example, where in the epidermis is Sun1/2 expressed? Is there any developmental regulation of LINC complex organization? One might predict reduced expression or altered localization of LINC complex components (such as SUN1/2) in the differentiated layers of the epidermis. Can the force-sensitive Lamin-A/C antibody be used in vivo to see if there are differences between staining patterns in the basal layer where cells have integrin-mediated contacts compared to the suprabasal, differentiating layers where they do not? Are there differences in nuclear morphology in the presence or absence of sun1/2 dKO epidermis that might support their model? Does loss of Sun1/Sun2 alter nuclear stiffness? If the authors can show that the regulation of molecular tension across the Linc complex directly correlates to changes in nuclear tension, this will strengthen the paper.

We performed immunostaining for SUN1 and SUN2 in E15.5 mouse skin. As described for post-natal skin in our prior manuscript (Stewart et al. JCB 2015), we find that both SUN1 and SUN2 are expressed throughout the basal and suprabasal layers of the epidermis (Figure 3—figure supplement 1A,B). We have been unable to perform staining for the force sensitive lamin-A/C antibody in skin sections due to poor antibody performance in IHC. Further, measuring nuclear stiffness directly, a technique with which we have experience, requires a substantial additional commitment and we have not attempted this approach here. We note, however, that the literature suggests that while LINC complexes are essential for force transduction to the nucleus, they have little effect on the force response (e.g. work from the Lammerding lab – Lombardi et al., JBC, 2011). We do not observe any gross changes in the nuclear morphology in *Sun* dKO epidermis, but would argue that our lack of understanding of the tension network in keratinocytes in vivo does not rule out there are nonetheless changes in tension exerted on the nucleus. Indeed, we have noted in our prior work that changes in nuclear shape can reflect both LINC complex-dependent and -independent factors (Stewart et al., JCB, 2015; Stewart et al., MBoC, 2019).

5. The molecular links how integrins control tension on the Linc complex, or how altered tension at the Linc complex controls differentiation are not at all addressed, and the paper should at least provide some insight into either upstream or downstream control of how integrin dependent regulation of Sun1/2 regulates differentiation; e.g. explore the molecular link between integrin and the linc complex (in their model both actin and MTs are potentially implicated for example), alterations in epigenetic marks, chromatin organization of e.g. relocalization of the EDC locus etc.The conditional b1 and KO of integrin a3 integrins do not result in altered/precocious differentiation in the skin (although it does affect proliferation and hair follicle morphogenesis). Presumably the link between integrins and the LINC complex under these conditions would be altered as well? The idea that LINC complex may control differentiation is possible, however the role of integrin based adhesions in this process has not been parsed out. At the very least the authors could have disrupted integrin signaling using KO cells or blocking antibodies. So it is possible that there may be alternative mechanisms? This should be considered.

We appreciate the reviewer’s feedback and agree that further investigation into the role of integrins in the regulation of LINC complex tension and keratinocyte differentiation is warranted. In this revision, we include new experiments to directly address the role that integrins play in regulating tension on the LINC complex by examining the tension on the N2G-JM-TSMod in β1 integrin KO MKCs. Our FRET analysis demonstrates that the FRET index of the N2G-TSMod was significantly higher in β1 integrin KO MKCs compared to WT MKCs (Figure 1J,K) indicating that β1 integrin is required for the high LINC complex tension state. Our additional data demonstrating that the N2G-TSMod is also sensitive to substrate stiffness (Figure 1F,G) further reinforces this relationship.

We also examined the expression of differentiation genes in β1 integrin KO MKCs. RT-qPCR analysis of *Sprr1b*, *Involucrin* (*Ivl*), and *S100a14* revealed a significantly elevated expression of these mRNAs in β1 integrin KO MKCs cultured in low calcium conditions (Figure 4I). These data mirror our observations in *Sun* dKO MKCs and support a role for β1 integrin in mediating tension on the LINC complex and its role in suppressing epidermal differentiation.

[Editors' note: further revisions were suggested prior to acceptance, as described below.]

We agree that your manuscript is much improved and has an important message for eLife. That said, there still several items remaining, that we feel need to be addressed prior to publication.Three issues remain. First, changes in nuclear structure and likely mechanics in keratinocytes induce rather global changes in chromatin and transcription (Le et al., 2016), and hence it would be helpful if you can present the ATAC data more generally and not just pick a few loci to analyze. Are your data specific for the EDC locus or are other differentiation genes also affected? One would also predict changes in the so-called stem cell genes-can you comment on whether this is observed? These minor adjustments will provide even a better global vision on how the nuclear envelope tension regulates gene expression and differentiation.

We have now added a supplemental panel (Figure 5—figure supplement 1B) and accompanying Supplementary File 4 to provide genome-wide analysis of the ATAC-seq data. Across the genome, most ATAC-seq peaks are found at the same genes for WT and Sun dKO MKCs. Sun dKO MKCs also show both gains and losses of ATAC-seq peaks associated with genes compared to WT MKCs, with a bias towards gains. Further analysis revealed that additional ATAC-seq peaks are found at genes encoding keratins and cell adhesion genes in Sun dKO MKCs, suggesting that epidermal differentiation genes outside of the EDC also show increased chromatin accessibility (Supplementary File 5). However, we do not see premature loss of ATAC-seq peaks at proliferation genes or “stem cell genes” (Supplementary File 5), arguing that precocious expression of epidermal differentiation genes is the primary alteration upon loss of the LINC complex. Of note, these data differ from the results of Le et al., 2016, who found global down-regulation of genes upon critical stretch of keratinocyte monolayers. We conclude, as outlined in the Discussion, that stretch achieved with the device employed by Le et al. and constitutive tension on the LINC complex dictated by integrin engagement represent two distinct contexts.

Second, the b1 KO cells have to be cultured in high calcium media and have very high levels of Rho activity, strong focal adhesions (nucleated through integrin b6) and massive stress fibers (Raghavan et al. 2003, Bandyopadhyay and Raghavan 2012). Given the role of the actin cytoskeletal network in transmitting the forces via the LINC complex to the nucleus, it seems surprising that the tension sensor shows such high FRET (low tension) compared to the WT cells. It would be helpful if you could add a bit of discussion as to how you interpret these data and reconcile this point. Toning down some of the conclusions made regarding this point might be warranted.

We appreciate this perspective. However, our data from this manuscript and other published work clearly indicate that high Rho activity and actin contractility are not sufficient to drive tension on the LINC complex – indeed this represents the critical point that lies at the heart of our model. For instance, cells at the interior of cohesive MKC colonies also display extensive stress fibers integrated at cell-cell adhesions (see our prior work – Mertz et al., PNAS, 2013; Stewart et al., JCB, 2015) and show low tension on the LINC complex and the nuclear lamina, as demonstrated in Figure 2 A-D. Thus, our data indicate that actomyosin contractility is not sufficient, in and of itself, to drive a high-tension state on the LINC complex. While we agree that these findings may be, on the surface, surprising, this model provides an explanation for how the LINC complex could specifically contribute to repressing epidermal differentiation genes in basal but not suprabasal keratinocytes. As for the specificity for integrin b1 and not integrin b6, further studies will be required to define why only engagement of the former is sufficient to exert high tension on the LINC complex. We now include a more detailed discussion on these points in the discussion of the revised manuscript.

Finally, it would seem that you should add a brief discussion and referencing of the recent Nava et al. Cell paper, that shows heterochromatin driven changes in nuclear mechanics in keratinocytes.

We now cite and discuss this paper, as suggested. However, we would point out that stretched cells and constitutive tension on LINC complexes with engaged b1-integrins likely represent distinct regimes as we do not observe the global changes observed by Le et al. and. Nava et al.